# Components of genetic associations across 2,138 phenotypes in the UK Biobank highlight adipocyte biology

Yosuke Tanigawa [1,14], Jiehan Li[2,3,14], Johanne M. Justesen[1,2,4], Heiko Horn [5,6], Matthew Aguirre [1,7], Christopher DeBoever[1,8], Chris Chang[9], Balasubramanian Narasimhan[1,10], Kasper Lage[5,6,11], Trevor Hastie[1,10], Chong Y. Park[2], Gill Bejerano [1,7,12,13], Erik Ingelsson [2,3,14] & Manuel A. Rivas [1,14]

Population-based biobanks with genomic and dense phenotype data provide opportunities for generating effective therapeutic hypotheses and understanding the genomic role in disease predisposition. To characterize latent components of genetic associations, we apply truncated singular value decomposition (DeGAs) to matrices of summary statistics derived from genome-wide association analyses across 2,138 phenotypes measured in 337,199 White British individuals in the UK Biobank study. We systematically identify key components of genetic associations and the contributions of variants, genes, and phenotypes to each component. As an illustration of the utility of the approach to inform downstream experiments, we report putative loss of function variants, rs114285050 (*GPR151*) and rs150090666 (*PDE3B*), that substantially contribute to obesity-related traits and experimentally demonstrate the role of these genes in adipocyte biology. Our approach to dissect components of genetic associations across the human phenome will accelerate biomedical hypothesis generation by providing insights on previously unexplored latent structures.

[1] Department of Biomedical Data Science, School of Medicine, Stanford University, Stanford, CA, USA. [2] Department of Medicine, Division of Cardiovascular Medicine, Stanford University, Stanford, CA, USA. [3] Stanford Cardiovascular Institute, Stanford University, Stanford, CA, USA. [4] Novo Nordisk Foundation Center for Basic Metabolic Research, Faculty of Health and Medical Sciences, University of Copenhagen, Copenhagen, Denmark. [5] Department of Surgery, Massachusetts General Hospital, Harvard Medical School, Boston, MA, USA. [6] Broad Institute of MIT and Harvard, Cambridge, MA, USA. [7] Department of Pediatrics, Stanford University School of Medicine, Stanford University, Stanford, CA, USA. [8] Department of Genetics, Stanford University, Stanford, CA, USA. [9] Grail, Inc., Menlo Park, CA, USA. [10] Department of Statistics, Stanford University, Stanford, CA, USA. [11] Institute for Biological Psychiatry, Mental Health Center Sct. Hans, University of Copenhagen, Roskilde, Denmark. [12] Department of Developmental Biology, Stanford University, Stanford, CA, USA. [13] Department of Computer Science, Stanford University, Stanford, CA, USA. [14]These authors contributed equally: Yosuke Tanigawa, Jiehan Li, Erik Ingelsson, Manuel A. Rivas. Correspondence and requests for materials should be addressed to E.I. (email: eriking@stanford.edu) or to M.A.R. (email: mrivas@stanford.edu)

Human genetic studies have been profoundly successful at identifying regions of the genome contributing to disease risk[1,2]. Despite these successes, there are challenges to translating findings to clinical advances, much due to the widespread pleiotropy and extreme polygenicity of complex traits, which are the presence of genetic effects of a variant across multiple phenotypes and multiple variants across a single phenotype[3–5]. In retrospect, this is not surprising given that most common diseases are multifactorial. However, it remains unclear exactly which factors, acting alone or in combination, contribute to disease risk and how those factors are shared across diseases. With the emergence of sequencing technologies, we are increasingly able to pinpoint alleles, possibly rare and with large effects, which may aid in therapeutic target prioritization[6–13]. Furthermore, large population-based biobanks, such as the UK Biobank, have aggregated data across tens of thousands of phenotypes[14]. Thus, an opportunity exists to characterize the phenome-wide landscape of genetic associations across the spectrum of genomic variation, from coding to non-coding, and rare to common.

Singular value decomposition (SVD), a mathematical approach developed by differential geometers[15], can be used to combine information from several (likely) correlated vectors to form basis vectors, which are guaranteed to be orthogonal and to explain maximum variance in the data, while preserving the linear structure that helps interpretation. In the field of human genetics, SVD is routinely employed to infer genetic population structure by calculating principal components using the genotype data of individuals[16].

To address the pervasive polygenicity and pleiotropy of complex traits, we propose an application of truncated SVD (TSVD), a reduced rank approximation of SVD[17–19], to characterize the underlying (latent) structure of genetic associations using summary statistics computed for 2,138 phenotypes measured in the UK Biobank population cohort[14]. We apply our approach, referred to as DeGAs — decomposition of genetic associations — to assess associations among latent components, phenotypes, variants, and genes. We highlight its application to body mass index (BMI), myocardial infarction (MI), and gallstones, motivated by high polygenicity in anthropometric traits, global burden, and economic costs, respectively. We assess the relevance of the inferred key components through GREAT genomic region

ontology enrichment analysis[20,21] and functional experiments. The results from DeGAs applied to protein-truncating variants (PTV) dataset indicate strong associations of targeted PTVs to obesity-related traits, while phenome-wide association analyses (PheWAS) uncover the differential region-specific regulation of our top candidates in fat deposition. For these reasons, we prioritize adipocytes as our experimental model system for the follow-up functional studies of our candidate genes. Given that the roles of adipocytes in regulating metabolic fitness have been established at both local and systemic levels of pathology associated with obesity, it is likely that the differentiation and function of adipocytes may shape the effects of our candidate genes at the cellular and molecular level.

## Results

**DeGAs method overview.** We generated summary statistics by performing genome-wide association studies (GWAS) of 2,138 phenotypes from the UK Biobank (Fig. 1a, Supplementary Table 1, Supplementary Data 1). We performed variant-level quality control, which includes linkage-disequilibrium (LD) pruning and removal of variants in the MHC region, to focus on 235,907 variants for subsequent analyses. Given the immediate biological consequence, subsequent downstream implications, and medical relevance of coding variants and predicted protein-truncating variants (PTVs), commonly referred to as loss-of-function variants[12,22,23], we performed separate analyses on three variant sets: (1) all directly-genotyped variants, (2) coding variants, and (3) PTVs (Supplementary Fig. 1). To eliminate unreliable estimates of genetic associations, we selected associations with p-values < 0.001, and standard error of beta value or log odds ratio of less than 0.08 and 0.2, respectively, for each dataset. The Z-scores of these associations were aggregated into a genome- and phenome-wide association summary statistic matrix $W$ of size $N \times M$, where $N$ and $M$ denote the number of phenotypes and variants, respectively. $N$ and $M$ were 2,138 and 235,907 for the "all" variant group; 2,064 and 16,135 for the "coding" variant group; and 628 and 784 for the PTV group. The rows and columns of $W$ correspond to the GWAS summary statistics of a phenotype and the phenome-wide association study (PheWAS) of a variant, respectively. Given its computational efficiency

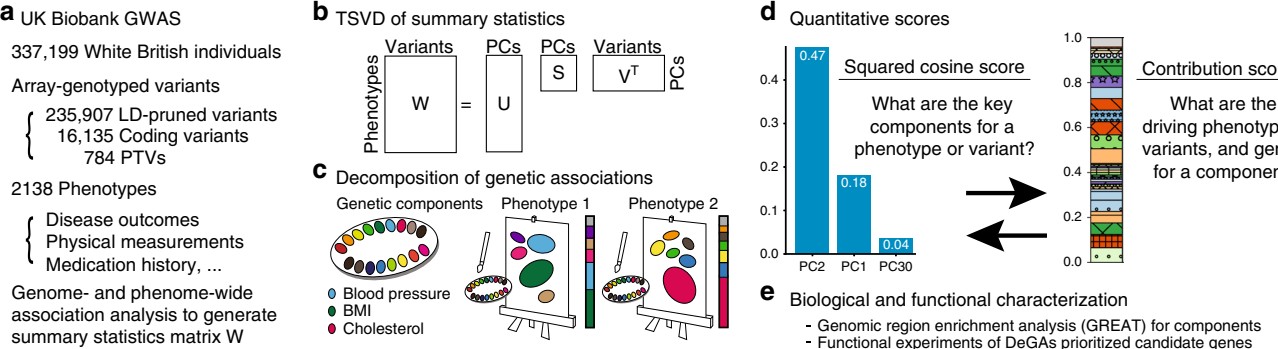

**Fig. 1** Illustrative study overview. **a** Summary of the UK Biobank genotype and phenotype data used in the study. We included White British individuals and analyzed LD-pruned and quality-controlled variants in relation to 2,138 phenotypes with a minimum of 100 individuals as cases (binary phenotypes) or non-missing values (quantitative phenotypes) (Supplementary Table 1, Supplementary Data 1). **b** Truncated singular value decomposition (TSVD) applied to decompose genome- and-phenome-wide summary statistic matrix $W$ to characterize latent components. $U$, $S$, and $V$ represent resulting matrices of singular values and vectors. **c** Decomposition of Genetic Associations (DeGAs) characterizes latent genetic components, which are represented as different colors on the palette, with an unsupervised learning approach — TSVD, followed by identification of the key components for each phenotype of our interest (painting phenotypes with colors) and annotation of each of the components with driving phenotypes, variants, and genes (finding the meanings of colors). **d** We used the squared cosine score and the contribution score, to quantify compositions and biomedical relevance of latent components. **e** We applied the genomic region enrichment analysis tool (GREAT) for biological characterization of each component and performed functional experiments focusing on adipocyte biology

compared to the vanilla SVD, we applied TSVD to each matrix and obtained a decomposition into three matrices $\mathbf{W} = \mathbf{USV}^T$ (**U**: phenotype, **S**: variance, **V**: variant). This reduced representation of $K = 100$ components altogether explained 41.9% (all), 62.8% (coding) and 75.5% (PTVs) of the variance in the original summary statistic matrices (Fig. 1b, Methods, Supplementary Fig. 2).

In DeGAs framework, we employ these latent components characterized from a densely phenotyped population-based cohort to investigate the genetics of common complex traits (Fig. 1c). To characterize each latent component and identify the relevant component given a phenotype, a gene, or a variant, or vice versa, we defined five different quantitative scores: phenotype squared cosine score, phenotype and variant contribution score, variant contribution score, and gene contribution score. The squared cosine scores quantify the relative importance of component for a given phenotype or gene, and are defined based on the squared distance of a component from the origin on the latent space. Contribution scores quantify relative importance of a phenotype, variant, or gene to a given component and is defined based on the squared distance of a phenotype, variant, or gene from the origin[24] (Fig. 1d, Methods). Using scores, DeGAs identifies the key latent components for a given complex trait and annotated them with the driving phenotypes, genes, and variants (Fig. 1c, Methods). We performed biological characterization of DeGAs components with the genomic region enrichment analysis tool (GREAT)[20,21] followed by functional experiments in adipocytes (Fig. 1e).

**Characterization of DeGAs latent components**. The PCA plots show the projection of phenotypes and variants onto DeGAs latent components. (Fig. 2a, b). For the variant PCA plot, we overlay biplot annotation as arrows to interpret the direction of the components (Fig. 2b, Methods). Overall, we find that the first five DeGAs components can be attributed to: 1) fat-free mass that accounts for the "healthy part" of body weight[25] (32.7%, Supplementary Table 2) and two intronic variants in *FTO* (rs17817449: contribution score of 1.15% to PC1, rs7187961: 0.41%); and a genetic variant proximal to *AC105393.1* (rs62106258: 0.46%); 2) fat mass and percentage measurements (61.5%) and the same three *FTO* and *AC105393.1* variants (rs17817449: 0.97%, rs7187961: 0.28%, rs62106258: 0.27%); 3) bioelectrical impedance measurements (38.7%), a standard method to estimate body fat percentage[26,27], and genetic variants proximal to *ACAN* (rs3817428: 0.64%), *ADAMTS3* (rs11729800: 0.31%), and *ADAMTS17* (rs72770234: 0.29%); 4) eye meridian measurements (80.9%), and two intronic variants in *WNT7B* (rs9330813: 5.73%, rs9330802: 1.14%) and a genetic variant proximal to *ATXN2* (rs653178: 0.96%); and 5) bioelectrical impedance and spirometry measures (45.4% and 26.0%, respectively) and genetic variants proximal to *FTO* (rs17817449: 0.17%), *ADAMTS3* (rs11729800: 0.11%), and *PSMC5* (rs13030: 0.11%) (Fig. 2c, d, Supplementary Data 2).

To highlight the ability of DeGAs to capture related sets of phenotypes, genes, and variants in genetic associations, we also applied TSVD to the missing-value imputed and Z-score transformed phenotype matrix and characterized the first 100 latent components (Methods). Using the individual and phenotype PCA plots, we found a fewer number of components that explains most of the variance and several phenotypes, such as traffic intensity of the nearest major road and creatinine (enzymatic) in urine, are dominantly driving the top phenotypic PCs (Supplementary Figs. 4, 5). We applied GWAS for each of the decomposed phenotypes (Supplementary Fig. 6). Through the genetic correlation analysis with the derived summary statistics, we found non-zero genetic correlations among the phenotypic PCs (Supplementary Figs. 7, 8).

**DeGAs application to BMI, MI, and gallstones**. To illustrate the application of DeGAs in characterizing the genetics of complex traits, we selected three phenotypes, BMI, MI, and gallstones, given the large contribution of anthropometric traits on the first five components, that ischemic heart diseases is a leading global fatal and non-fatal burden, and that gallstones is a common condition with severe pain and large economic costs where polygenic risk factors are largely unknown[28,29]. We identified the top three key components for these three phenotypes with DeGAs using the "all" variants dataset.

For BMI, we find that the top three components of genetic associations (PC2, PC1, and PC30) altogether explained over 69% of the genetic associations (47, 18, and 4%, respectively, Supplementary Fig. 3a). The top two components (PC2 and PC1) corresponded to components of body fat (PC2) and fat-free mass measures (PC1), as described above. PC30 was driven by fat mass (28.7%) and fat-free mass (6.8%), but also by non-melanoma skin cancer (7.72%) — linked to BMI in epidemiological studies[30] — and childhood sunburn (7.61%) (Fig. 3a, Supplementary Data 2).

For MI, a complex disease influenced by multiple risk factors[31], we found that the top components were attributed to genetics of lipid metabolism (PC22, high-cholesterol, statin intake, and *APOC1*), alcohol intake (PC100), and sleep duration and food intake (PC83, 25.2%) that collectively corresponded to 36% of the genetic associations (Fig. 3a, Supplementary Figs. 3b, 9, 10, Supplementary Data 2).

Cholelithiasis is a disease involving the presence of gallstones, which are concretions that form in the biliary tract, usually in the gallbladder[32]. We found that the top components contributing to gallstones corresponded to associations with fresh fruit (PC72) and water intake (PC64), as well as bioelectrical impedance of whole body (PC67) corresponding to 51% of genetic associations altogether (Fig. 3a, Supplementary Figs. 3c, 9, 11, Supplementary Data 2). We confirmed the robustness of these results with respect to the selection of number of components, $K$ (Methods, Supplementary Figs. 12–16).

**Biological characterization of DeGAs components**. To provide biological characterization of the key components, we applied the genomic region enrichment analysis tool (GREAT)[20,21] to dissect the biological relevance of the identified components with both coding and non-coding variants. Given the coverage of the manually curated knowledge of mammalian phenotypes, we focused on the mouse genome informatics (MGI) phenotype ontology and set $p = 5 \times 10^{-6}$ as the Bonferroni-corrected statistical significance threshold (Method)[33]. For each key component, we applied GREAT and found an enrichment for the mouse phenotypes consistent with the phenotypic description of our diseases of interest[20,21]. The top component for BMI, identified as the body fat measures component (PC2), showed enrichment of several anthropometric terms including abnormally short feet (brachypodia) (MP:0002772, binomial fold = 9.04, $p = 1.3 \times 10^{-23}$), increased birth weight (MP:0009673, fold = 6.21, $p = 1.3 \times 10^{-11}$), and increased body length (MP:0001257, binomial fold = 3.01, $p = 1.3 \times 10^{-36}$) (Fig. 3b, Supplementary Data 3). For MI, we found enrichment of cardiac terms, such as artery occlusion (PC22, MP:0006134, fold = 15.86, $p = 1.14 \times 10^{-25}$) and aortitis (PC22, MP:0010139, aorta inflammation, fold = 9.36, $p = 3.41 \times 10^{-31}$) (Supplementary Fig. 17, Supplementary Data 4). Similarly, for gallstones, the top enrichment was for abnormal circulating phytosterol level (PC72, MP:0010075, fold = 11.54, $p = 5.51 \times 10^{-11}$), which is known to be involved in gallstone development[34] (Supplementary Fig. 18, Supplementary Data 5).

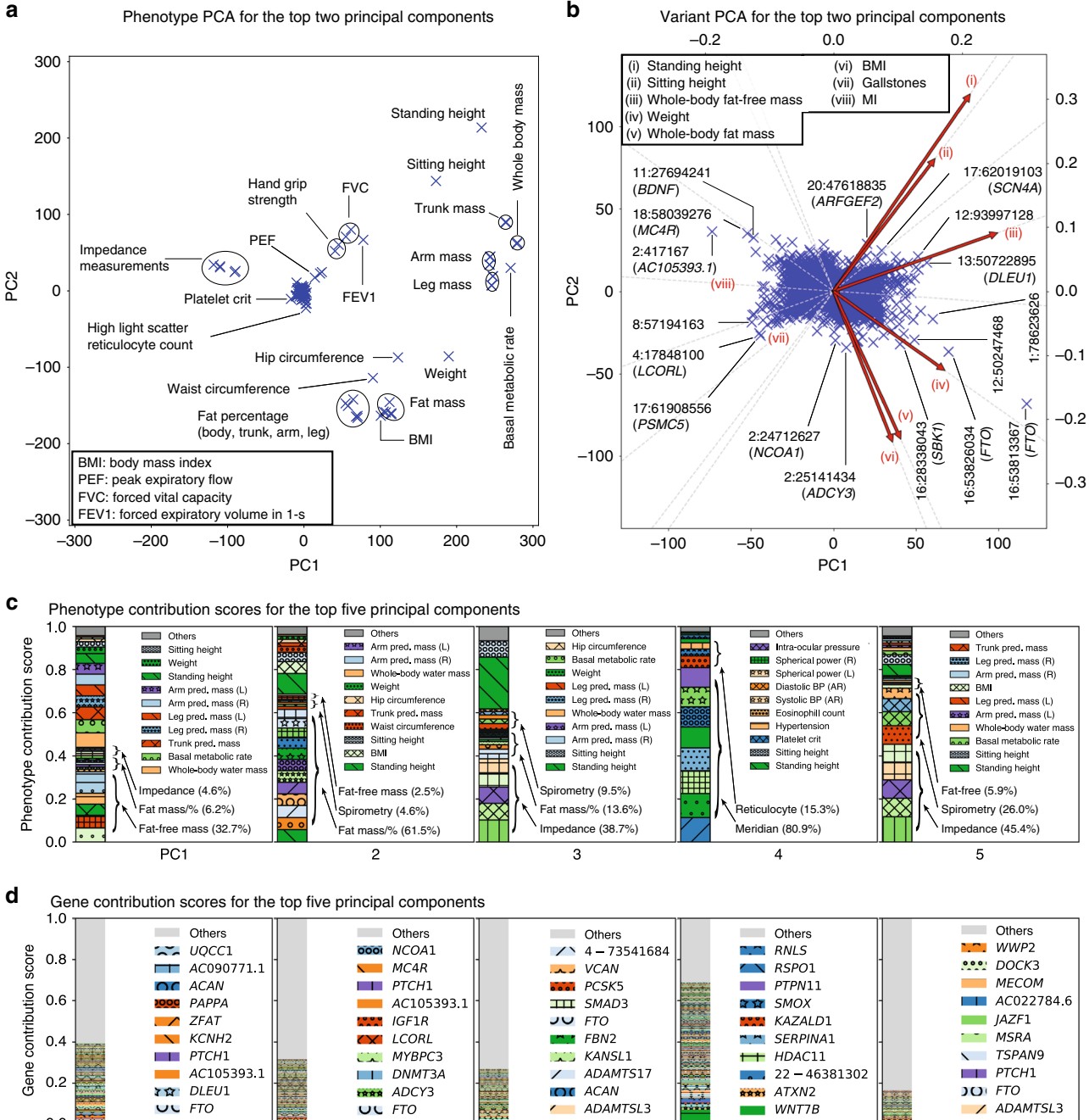

**Fig. 2** Characterization of DeGAs latent structures. **a**, **b** Components from truncated singular value decomposition (TSVD) corresponds to principal components in the phenotype (**a**) and variant (**b**) spaces. The first two components of all the variants, excluding the MHC region, and relevant phenotypes are shown. **b** For variant PCA, we show biplot arrows (red) for selected phenotypes to help interpretation of the direction of principal components (Methods). The variants are labeled based on the genomic positions and the corresponding gene symbols. For example, "16:53813367 (*FTO*)" indicates the variant in gene *FTO* at position 53813367 on chromosome 16. **c**, **d** Phenotype (**c**) and gene (**d**) contribution scores for the first five components. PC1 is driven by largest part of the body mass that accounts for the "healthy part" (main text) including whole-body fat-free mass and genetic variants on *FTO* and *DLEU1*, whereas PC2 is driven by fat-related measurements, PC3 is driven by bioelectrical impedance measurements, PC4 is driven by eye measurements, and PC5 is driven by bioelectrical impedance and spirometry measurements along with the corresponding genetic variants (main text, Supplementary Table 2, Supplementary Data 2). Each colored segment represents a phenotype or gene with at least 0.5% and 0.05% of phenotype and gene contribution scores, respectively, and the rest is aggregated as others on the top of the stacked bar plots. The major contributing phenotype groups (Methods, Supplementary Table 2) and additional top 10 phenotypes and the top 10 genes for each component are annotated in **c** and **d**, respectively. pred.: predicted, %: percentage, mass/% mass and percentage, BP: blood pressure, AR: automated reading, L: left, R: right. Source data are provided as a Source Data file (**a**, **b**) and in Supplementary Data 2 (**c**, **d**)

**a** Driving phenotypes of the top three key components for body mass index (BMI), myocardial infarction (MI), and gallstones

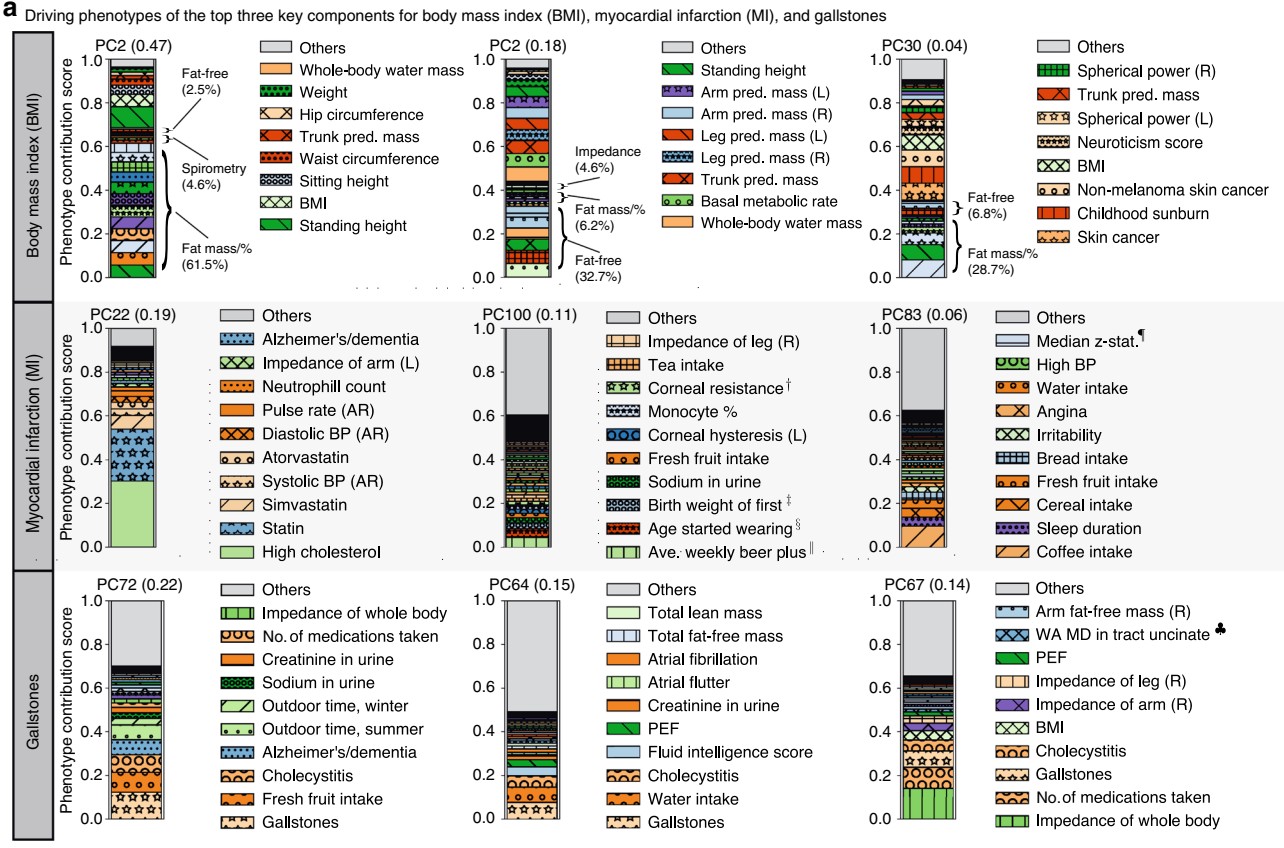

**b** Ontology enrichment analysis with the genomic region enrichment analysis tool (GREAT) for body mass index

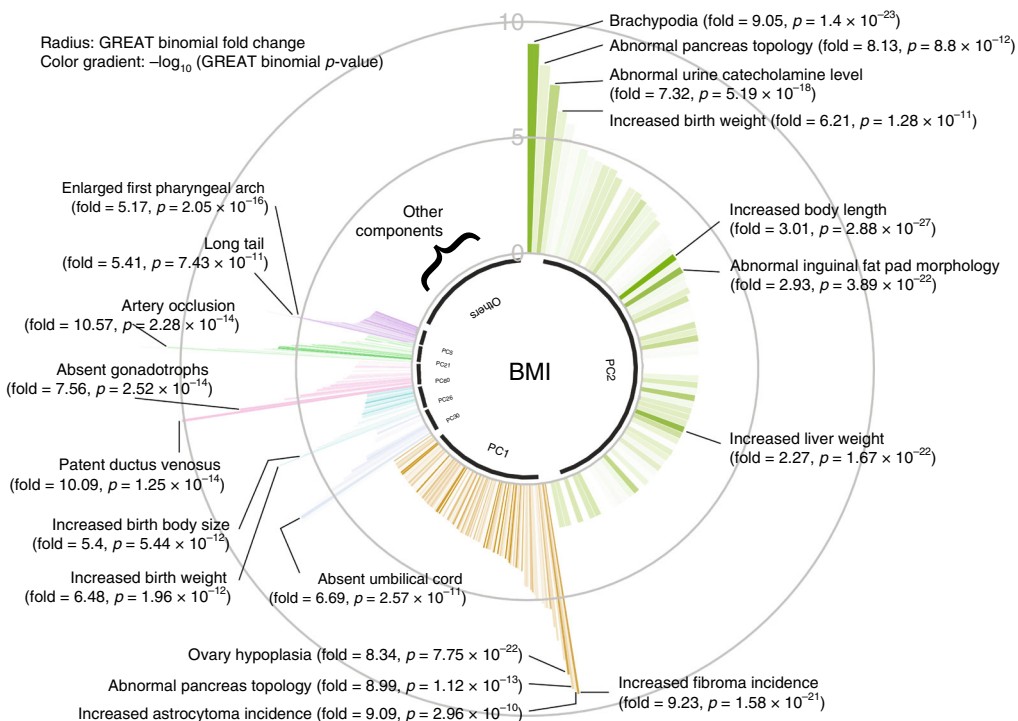

To test the specificity of the enriched ontology terms while considering the correlation structure within ontology terms, we took the top five enriched terms for each DeGAs component, obtained the list of genes annotated with these top terms, and measured their pairwise gene set similarity across 100 DeGAs components using Jaccard index (Methods). Jaccard index is a set similarity measure ranges between zero and one, where one means the complete match and zero means complete disjoint of the two sets. We found the median of the pairwise similarity to be 0.029 (Supplementary Fig. 19).

**Fig. 3** The top three key DeGAs components for BMI, MI, and gallstones. **a** The top three key components for each phenotype are identified by phenotype squared cosine scores and characterized with the driving phenotypes by phenotype contribution scores (Methods). Each colored segment represents a phenotype with at least 0.5% of phenotype contribution scores for each of the component and the rest of the phenotypes are aggregated as others and shown as the gray bar on the top. For BMI, additional phenotype grouping is applied (Methods, Supplementary Table 2). **b** Biological characterization of driving non-coding and coding variants of the key components for BMI with GREAT. The key components are shown proportional to their squared cosine score along with significantly enriched terms in mouse MGI phenotype ontology. The radius represents binomial fold change and the color gradient represents $-\log_{10}(p\text{-value})$ from GREAT ontology enrichment analysis. pred.: predicted, #: number, %: percentage, mass/% mass and percentage, BP: blood pressure, AR: automated reading, L: left, R: right, WA: weighted average. †: Corneal resistance factor (right), ‡: Birth weight of first child, §: Age started wearing glasses or contact lenses, ||: Average weekly beer plus cider intake, ¶: Median z-statistic (in group-defined mask) for shapes activation, ♣: Weighted-mean MD in tract uncinate fasciculus (right). Source data are provided in Supplementary Data 2-3

**Coding and protein-truncating variants**. Given the challenges with interpreting genetic associations across thousands of possibly correlated phenotypes and diverse variant functional categories, we applied DeGAs to coding variant-phenotype associations and PTV associations. For the coding dataset, we identified PC2 and PC1 as the top two key components for BMI, with 51% and 14% of phenotype squared contribution scores, respectively (Supplementary Fig. 20). The major drivers of these two components include fat mass measurements (55.2% of phenotype contribution score for PC2), fat-free mass measurements (33.3%, PC1), genetic variants on *MC4R* (3.7% gene contribution score for PC2), and *ZFAT* (3.4% gene contribution score for PC1) (Supplementary Figs. 21–22, Supplementary Data 2).

Predicted PTVs are a special class of protein-coding genetic variants with possibly strong effects on gene function[9,12,22,35]. More importantly, strong effect PTV-trait associations can uncover promising drug targets, especially when the direction of effect is consistent with protection of human disease. Using the PTV dataset, we identified PC1 and PC3 as the top two key components for BMI, with 28% and 12% of phenotype squared contribution scores, respectively (Supplementary Fig. 23). The major drivers of PC1 were weight-related measurements, including left and right leg fat-free mass (5.0% and 3.7% of phenotype contribution score for PC1, respectively), left and right leg predicted mass (4.9% each), weight (4.6%), and basal metabolic rate (4.6%), whereas the drivers of PC3 included standing height (13.7%), sitting height (8.1%), and high reticulocyte percentage (6.4%) (Fig. 4a, Supplementary Data 2). Top contributing PTVs to PC1 included variants in *PDE3B* (19.0%), *GPR151* (12.3%), and *ABTB1* (8.5%), whereas PC3 was driven by PTVs on *TMEM91* (8.6%), *EML2-AS1* (6.7%), and *KIAA0586* (6.0%) (Fig. 4b, Supplementary Data 2).

Based on stop-gain variants in *GPR151* (rs114285050) and *PDE3B* (rs150090666) being key contributors to the top two components of genetic associations for PTVs and BMI (Fig. 4c), we proceeded to detailed phenome-wide association analysis (PheWAS) assessing associations of these PTVs with anthropometric phenotypes. PheWAS analysis of these variants confirmed strong associations with obesity-related phenotypes including waist circumference (*GPR151*, marginal association beta = $-0.065$, $p = 2.5 \times 10^{-8}$), whole-body fat mass (*GPR151*, beta = $-0.069$, $p = 1.4 \times 10^{-7}$), trunk fat mass (*GPR151*, beta = $-0.071$, $p = 1.5 \times 10^{-7}$), hip circumference (*PDE3B*, beta = 0.248, $p = 1.8 \times 10^{-11}$), right leg fat-free mass (*PDE3B*, beta = 0.129, $p = 4.2 \times 10^{-8}$) and body weight (*PDE3B*, beta = 0.177, $p = 4.6 \times 10^{-8}$) (Fig. 4d, Supplementary Fig. 24, Supplementary Tables 3-4). Among 337,199 White British individuals, we found 7,560 heterozygous and 36 homozygous carriers of the *GPR151* variant and 947 heterozygous carriers of *PDE3B* variants. To assess the effect of the PTVs on BMI, a commonly-used measure of obesity, we performed univariate linear regression analysis with age, sex, and the first four genetic PCs as covariates and found that heterozygous and carriers of *GPR151* PTVs showed 0.324 kg m$^{-2}$

lower BMI than the average UK Biobank participant ($p = 4.13 \times 10^{-7}$). We did not find evidence of association with homozygous carriers ($n = 28$; $p = 0.665$), presumably due to lack of power (Supplementary Fig. 25). Heterozygous carriers of *PDE3B* PTVs showed 0.647 kg m$^{-2}$ higher BMI ($p = 2.09 \times 10^{-4}$) than the average UK Biobank participant (Supplementary Fig. 26).

**Functional follow-up of candidate genes in adipocyte models**. We sought to illustrate the potential application of DeGAs in prioritizing candidate genes using functional follow-up experiments. Several of our most interesting findings were observed from strong associations between PTVs and obesity-related traits. Variants in *GPR151* and *PDE3B* are the two strongest contributors, albeit in opposite directions, to the top component (PC1) driving the genetic associations between PTVs and BMI (Fig. 4a–c). In addition to BMI, a simple indicator of overall body fat level, PheWAS studies have suggested strong correlations between regional body fat distribution and these two PTVs, with *GPR151* being more considerably associated with waist circumference and trunk fat (Fig. 4d), while *PDE3B* was more notably related to hip circumference and lower-body fat (Supplementary Fig. 24). Regional fat deposition is more accurately reflected by the local development and function of adipocytes in terms of size, number and lipid content. In order to explore how these two candidates regulate body fat composition differently, we chose to study their impacts on biological characteristics of adipocytes. Specifically, the expression and function of *PDE3B* and *GPR151* were evaluated in mouse 3T3-L1 and human Simpson-Golabi-Behmel Syndrome (SGBS) cells, two well-established preadipocyte models used for studying adipocyte differentiation (i.e. adipogenesis) and function[36,37].

First, we demonstrated that both genes were expressed in preadipocytes, but showed different expression patterns when cells were transforming into mature adipocytes: *PDE3B* increased dramatically during both mouse and human adipogenesis, while *GPR151* maintained a low expression level throughout the differentiation (Fig. 5a, b). Next, to explore the causal relationships between gene expression and adipogenesis, we introduced short interfering RNA (siRNA) against *Pde3b* and *Gpr151*, respectively, into 3T3-L1 preadipocytes and monitored the impact of gene knockdown on conversion of preadipocytes to adipocytes. Knockdown of *Gpr151*, by 3 siRNAs individually and together (Fig. 5c), drastically impaired adipocyte differentiation, as evidenced by lowered expression of adipogenesis markers (*Pparg, Cebpa,* and *Fabp4*) (Fig. 5d), as well as the reduced formation of lipid-containing adipocytes (Fig. 5e, f). Further, to test the functional capacity of the fat cells lacking *Gpr151*, we performed a lipolysis assay — an essential metabolic pathway of adipocytes and thus, a key indicator of adipocyte function — on mature adipocytes derived from preadipocytes transfected with either scrambled siRNA (scRNA) or si*Gpr151*. Not surprisingly, *Gpr151*-deficient lipid-poor adipocytes showed dramatically lower lipolysis, along with the impaired capability of responding

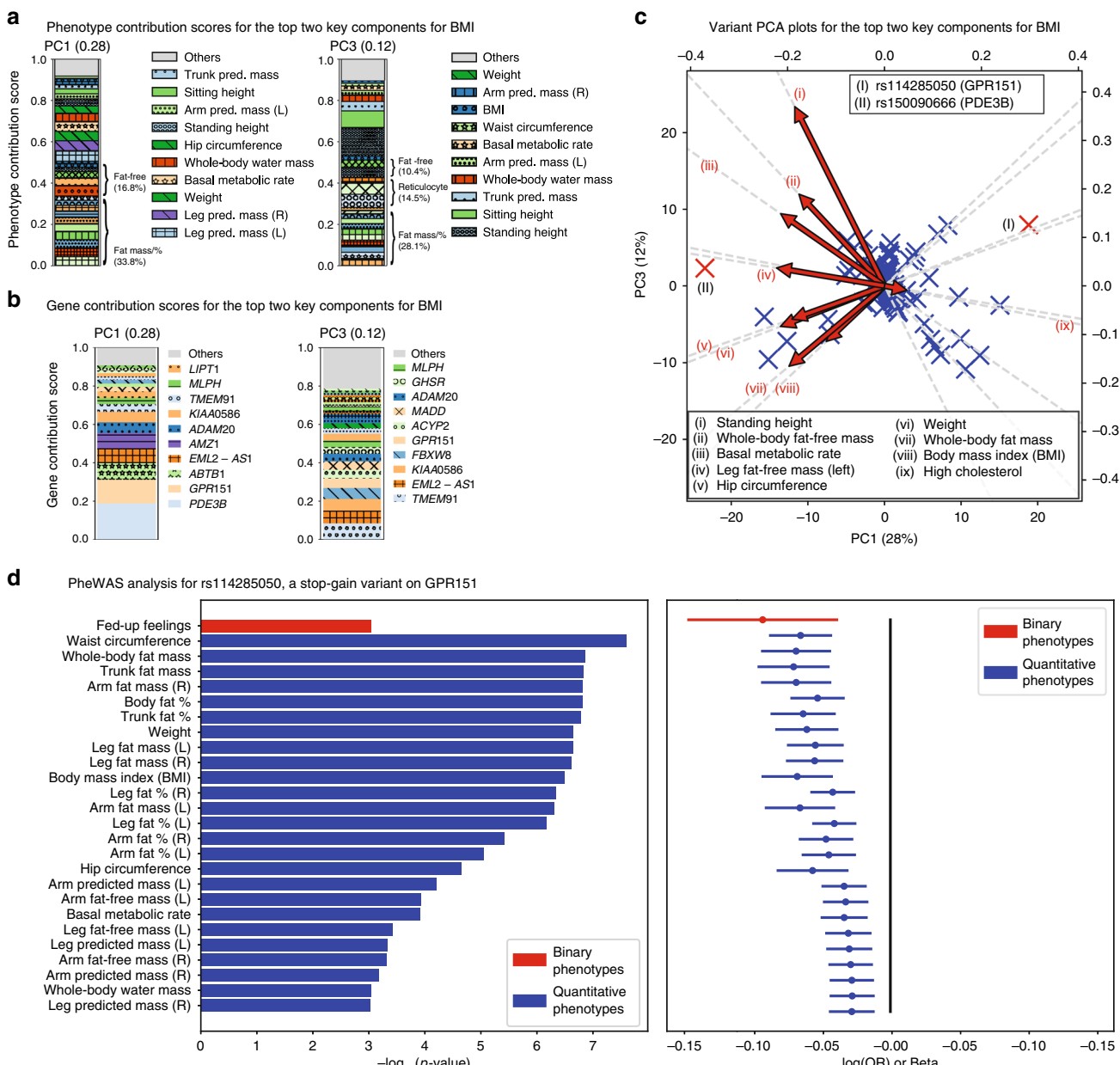

**Fig. 4** DeGAs applied to the protein-truncating variants (PTVs) dataset. **a**, **b** Phenotype (**a**) and gene (**b**) contribution scores for the top key components associated with BMI based on phenotype grouping (Methods, Supplementary Table 2). **c** Variant PCA plot with biplot annotations for the top two components (Methods). The identified targets for functional follow-up (main text) are marked as (I) rs114285050 (a stop-gain variant on *GPR151*) and (II) rs150090666 (*PDE3B*). **d** Phenome-wide association analysis for *GPR151* rs114285050. The *p*-values (left) and log odds ratio (OR) (binary phenotypes, shown as red) or beta (quantitative phenotypes, shown as blue) (right) along with 95% confidence interval are shown for the phenotypes with minimum case count of 1000 (binary phenotypes) or 1000 individuals with non-missing values (quantitative phenotypes) and strong association (*p* < 0.001) and with this variants among all the phenotypes used in the study (*n* = 337,199 White British individuals in the UK Biobank for binary traits and *n* > 331,000 for each quantitative trait, Supplementary Table 3). L: left, R: right, %: percentage, pred: predicted. Source data are provided in Supplementary Data 2 (**a**, **b**), Source Data file (**c**), and Supplementary Table 3 (**d**)

to isoproterenol (ISO), a β-adrenergic stimulus of lipolysis (Fig. 5g). These data suggest that *GPR151* knockdown in adipocyte progenitor cells may block their conversion into mature adipocytes.

To further analyze the functional impact of GPR151 in adipocytes, we generated an overexpression model of GPR151 by infecting 3T3-L1 preadipocytes with virus expressing Flag-tagged human *GPR151* driven by either EF1α or aP2 promotor (Supplementary Fig. 27a). Overexpression of *GPR151* by both constructs were confirmed at the gene and protein levels

(Supplementary Fig. 27b–d). However, despite the substantial effect of *Gpr151* knockdown on adipogenesis (Fig. 5), over-expression of *GPR151* in preadipocytes failed to influence adipocyte differentiation significantly, as shown by similar levels of adipogenic markers compared to the non-infected controls (Supplementary Fig. 27e, f). To eliminate the potential masking effects of any unperturbed cells in the partially infected cell population, we specifically selected *GPR151*-overexpressing cells by staining Flag-GPR151 positive cells with APC-conjugated flag antibody and sorted APC+ and APC− cells from the

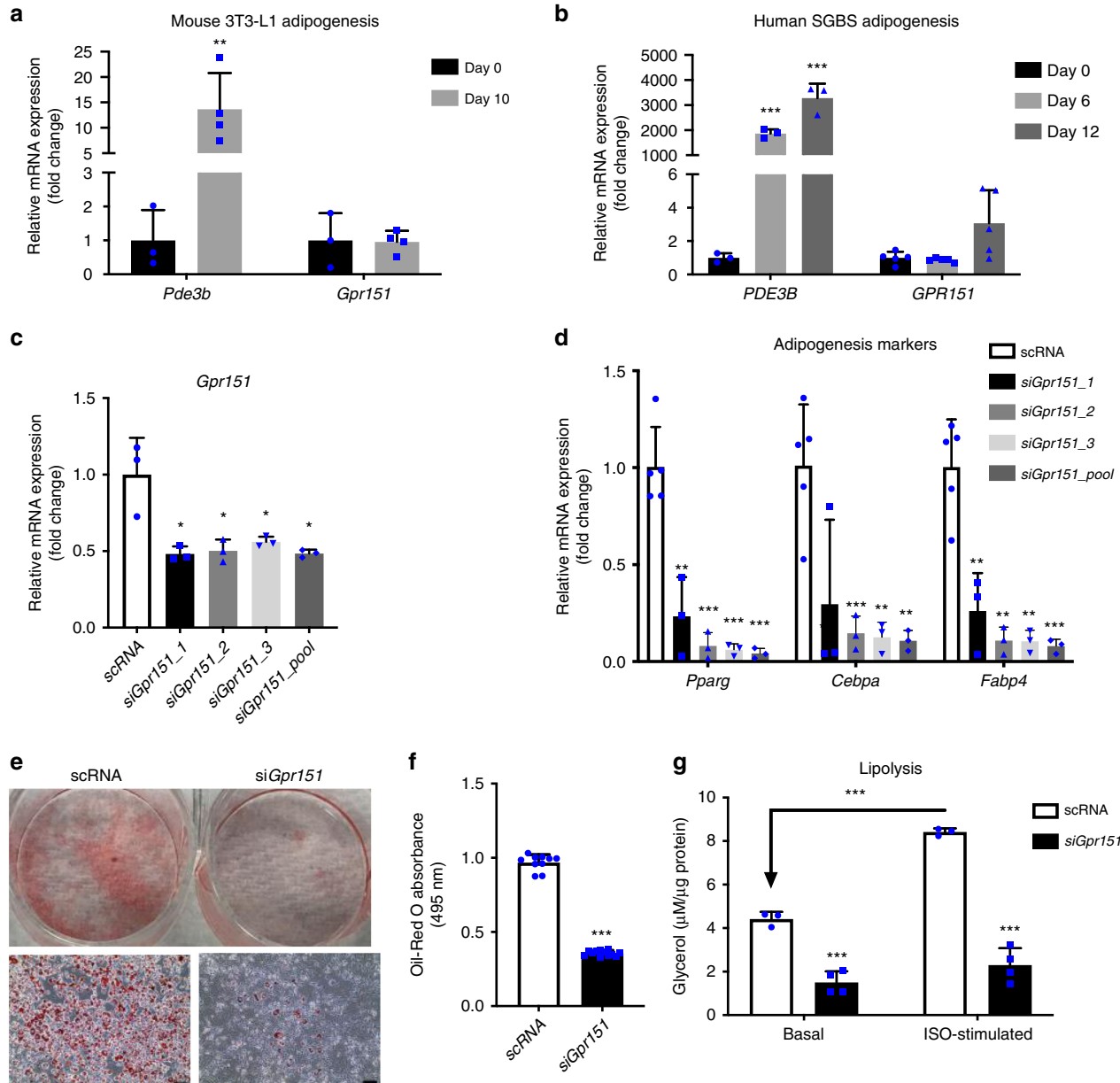

**Fig. 5** Experimental validation of *GPR151* and *PDE3B* function in adipogenesis. **a**, **b** qPCR analysis of gene expression patterns of *PDE3B* and *GPR151* during (**a**) mouse 3T3-L1 adipogenesis and (**b**) human SGBS adipogenesis. For 3T3-L1 cells, $n = 3$ independent samples for day 0, $n = 4$ for day 10. For SGBS cells, $n = 3$ independent samples for all timepoints of *PDE3B* analysis, $n = 5$ for all timepoints of *GPR151* analysis. **c** qPCR analysis of *Gpr151* mRNA knockdown in 3T3-L1 preadipocytes, by 3 siRNAs targeting *Gpr151* individually and together. $n = 3$ independent experiments. **d** qPCR analysis of the effect of si*Gpr151* knockdown (individually and together) on adipogenesis markers, *Pparg*, *Cebpa* and *Fabp4*. $n = 3$ independent experiments. **e**–**g** Oil-Red O staining (**e**), quantification of lipid droplets (**f**), and lipolysis (**g**) in scRNA- or si*Gpr151*-tansfected adipocytes. ×10 magnification. Scale bar = 100 μm. For ORO imaging and quantification, $n = 10$ independent cell cultures for scRNA, $n = 12$ for si*Gpr151*. For lipolysis, $n = 3$ independent experiments. Means ± SEM are shown (***$p$-value < 0.001, **$p$-value < 0.01, *$p$-value < 0.05). scRNA: scrambled siRNA. ISO: isoproterenol. Source data are provided as a Source Data file

differentiating adipocyte cultures (Supplementary Fig. 27g–l). In both EF1α- and aP2-driven *GPR151* overexpression models, *GPR151* mRNA levels were enriched in APC+ cells compared to APC− cells (Supplementary Fig. 27m–n). However, APC+ cells expressed genes characteristics of differentiating adipocytes in a similar level to that of APC− cells (Supplementary Fig. 27m–n). These data conclude that overexpression of GPR151 in preadipocytes cannot further enhance adipogenesis, suggesting that the endogenous level of GPR151 in preadipocytes may be sufficient to maintain the normal differentiation potential of preadipocytes. Although GPR151 is predominantly expressed in the brain, especially in hypothalamic neurons that control

appetite and energy expenditure[38], we identified that the GPR151 protein is present in both subcutaneous and visceral adipose tissue from mice (SAT and VAT), albeit in a very low level (Supplementary Fig. 27o). To sum up the results from the gain- and loss-of-function studies of *GPR151* in preadipocyte models, minimal but indispensable endogenous expression of GPR151 in adipose progenitor cells in generating lipid-rich adipocytes may underlie one of the mechanisms by which GPR151 promotes obesity.

In contrast to *GPR151*, knockdown of *Pde3b* in 3T3-L1 preadipocytes (Supplementary Fig. 28a) showed no significant influence on adipogenesis and lipolysis (under either basal or β-

adrenergic stimulated conditions), as compared to scRNA-transfected controls (Supplementary Fig. 28b–e). Since *PDE3B* is expressed primarily in differentiated adipocytes (Fig. 5a, b), future research efforts should be concentrated on studying the metabolic role of PDE3B in mature adipocytes.

Collectively, we performed functional characterization studies on the top two genes contributing to obesity-related traits, as selected based on the DeGAs approach in an unbiased manner, in hopes of (1) providing an example of how to interpret the results from DeGAs effectively and how to select candidate genes for the relevant experimental models; and (2) initiating an application of DeGAs in biological research and inspiring more state-of-art translational studies of candidates predicted from DeGAs.

## Discussion

We developed DeGAs, an application of TSVD, to decompose genome- and phenome-wide summary statistic matrix from association studies of thousands of phenotypes for systematic characterization of latent components of genetic associations and advanced the understanding on polygenic and pleiotropic architecture of complex traits. Applying DeGAs, we identified key latent components characterized with disease outcomes, risk factors, comorbidity structures, and environmental factors, with corresponding sets of genes and variants, providing insights on their context-specific functions. We demonstrated the robustness of the results by applying DeGAs with different parameters. With additional biological characterization of latent components using GREAT, we find component-specific enrichment of relevant phenotypes in mouse phenotype ontology. This replication across species highlights the ability of DeGAs to capture functionally relevant sets of both coding and non-coding variants in each component.

Our comparison of DeGAs to an alternative approach — decomposition of individual phenotype data followed by GWAS — highlights the ability of DeGAs to capture most of the variation in the genetic associations and to enable identification of bio-medically relevant genetic signals as latent components connecting sets of genetic variants and phenotypes. As an illustration of in-depth analysis of genetic variants with different functional consequences, we reported applications of DeGAs for different functional categories.

In DeGAs, we provided multiple ways to investigate the biological relevance of latent components, including quantitative scores and ontology enrichment analysis. These metrics are useful to annotate and interpret latent components, which are otherwise just mathematical objects in a high-dimensional space. For example, we found a significant contribution of anthropometric traits among the top 5 components, which reflects the pervasive polygenicity of these traits (Supplementary Fig. 29)[39,40]. By leveraging the ability of TSVD to efficiently summarize most of the variance in the input association statistic matrix, DeGAs provides a systematic way to interpret polygenic and pleiotropic genetic architecture of common complex traits.

Given that DeGAs is applied to summary statistics and does not require individual-level data, there is substantial potential to dissect genetic components of the human phenome when applied to data from population-based biobanks around the globe[14,41–44]. In fact, we are the first to develop a computational method that can jointly analyze genetics of thousands of phenotypes from a densely phenotyped population. As a proof of concept, we demonstrate a strategy to identify candidate genes for translational studies for obesity or its complications based on combination of quantitative results from DeGAs, phenome-wide analyses in the UK Biobank, and functional studies in adipocytes. Due to differences in phenotype and variant selection, it is

possible that the latent structures discovered from DeGAs would be different if using GWAS summary statistics from a different GWAS study. However, DeGAs is capable of identifying the most relevant components for a given input dataset using quantitative scores. In fact, our analysis for the three datasets — "all", coding, and PTVs — identified different PCs for each trait of our interest, but their characterization with contribution scores enabled interpretation of the DeGAs components.

To select candidate genes and their most relevant experimental models for the functional studies in an unbiased manner, we applied DeGAs to PTVs dataset. First, we identified PC1 as the top phenotype contribution component to BMI. Second, based on the fact that the main drivers of PC1 phenotypes were fat mass-related measurements, we chose adipocytes as our experimental model to approach candidate gene studies. Last but not least, we unbiasedly selected the top two genes contributing PTVs to PC1 (variants in *GPR151* and *PDE3B*) and explored their functionality in fat cells, in order to illustrate the application of DeGAs computational analysis in biological research.

As a result, loss-of-function of *GPR151* in adipocyte precursors limited their conversion into lipid-rich adipocytes. This result is directionally consistent with our DeGAs and univariate regression analysis showing that *GPR151* PTVs are associated with lower obesity and fat mass, especially central obesity (e.g., waist circumference and trunk fat mass) (Fig. 4d). GPCRs are known to influence adipogenesis by conveying a complex series of secondary messengers, including cAMP and calcium signals[45,46]. The density of receptors and the timing of receptor expression during adipogenesis governs the level, timing, and duration of the secondary signals, which is a critical factor in initiating and/or maintaining adipocyte conversion[45]. Although the endogenous expression of *GPR151* is low in preadipocytes, our results show that its presence is important to instigate the early events of adipogenic differentiation. Further investigation of the mechanism of *GPR151* action is valuable to understand its integral role in adipogenesis to the full extent.

*PDE3B*, on the other hand, did not affect differentiation of preadipocytes significantly in our study. There is evidence that *PDE3B* plays a more notable role in differentiated mature adipocytes, the primary component of adipose tissue. As an essential enzyme that hydrolyzes both cAMP and cGMP, *PDE3B* is known to be predominantly expressed in tissues that are important in regulating energy homeostasis, including adipose tissue[47]. White adipose tissue in *Pde3b* knockout mice behaves more as "beige" fat with improved mitochondrial activity and energy-burning properties, leading to a reduction of visceral fat mass as compared to the wild-type littermates[48]. Moreover, *Pde3b* knockout in mice confers cardioprotective effects[49], and human *PDE3B* "knockout" subjects display lower circulating triglycerides and higher HDL-cholesterol in blood[50]. There is a growing body of evidence that cardiometabolic health is linked to improved body fat distribution (i.e. lower visceral fat, higher subcutaneous fat)[51]. Our PheWAS analysis suggests that *PDE3B* PTVs have the strongest association to hip circumference (e.g., lower-body subcutaneous adiposity) (Supplementary Fig. 24). Therefore, understanding the fat depot-specific metabolic effects of *PDE3B* may help uncover the mechanism underlying the positive relationship of *PDE3B* PTVs with peripheral fat accumulation and favorable metabolic profiles.

Taken together, we highlight the directional concordance of our experimental data with the quantitative results from DeGAs and PTV-phenotype associations: *GPR151* inhibition may reduce total body and central fat, while deletion of *PDE3B* may favor subcutaneous, rather than visceral, fat deposition; both are expected to have beneficial effects on cardiometabolic health. Although these two genes were recently reported to be associated with obesity in another recent study based on the UK Biobank[52],

we are the first to experimentally identify a requisite role of *GPR151* in regulating adipogenesis. We also suggest a role of *PDE3B* in mature adipocytes to impose the cardioprotective effects of *PDE3B* inhibitors[49]. In this study, we focused on evaluating the functional effects of these genes on adipocyte function and development. We do not exclude the contribution nor the importance of other tissues or mechanisms underlying body weight changes. Indeed, some lines of evidence support additional effects of *GPR151* on obesity via the central nervous system — possibly on appetite regulation[38], while loss-of-function in *PDE3B* is also associated with height[52] — another contributing factor to body weight changes.

The resource made available with this study, including the DeGAs app, an interactive web application in the Global Biobank Engine[53], provides a starting point to investigate genetic components, their functional relevance, and potential therapeutic targets. These results highlight the benefit of comprehensive phenotyping on a population and suggest that systematic characterization and analysis of genetic associations across the human phenome will be an important part of efforts to understand biology and develop therapeutic approaches.

## Methods

**Compliance with ethical regulations and informed consent**. This research has been conducted using the UK Biobank Resource under Application Number 24983, "Generating effective therapeutic hypotheses from genomic and hospital linkage data" (http://www.ukbiobank.ac.uk/wp-content/uploads/2017/06/24983-Dr-Manuel-Rivas.pdf). Based on the information provided in Protocol 44532 the Stanford IRB has determined that the research does not involve human subjects as defined in 45 CFR 46.102(f) or 21 CFR 50.3(g). All participants of UK Biobank provided written informed consent (more information is available at https://www.ukbiobank.ac.uk/2018/02/gdpr/). All DNA samples and data in this study were pseudonymized.

**Study population**. The UK Biobank is a population-based cohort study collected from multiple sites across the United Kingdom. Information on genotyping and quality control has previously been described[14]. In brief, study participants were genotyped using two similar arrays (Applied Biosystems UK BiLEVE Axiom Array (807,411 markers) and the UK Biobank Axiom Array (825,927 markers)), which were designed for the UK Biobank study. The initial quality control was performed by the UK Biobank analysis team and designed to accommodate the large-scale dataset of ethnically diverse participants, genotyped in many batches, using two similar arrays[14].

**Genotype data preparation**. We used genotype data from the UK Biobank dataset release version 2[14] and the hg19 human genome reference for all analyses in the study. To minimize the variabilities due to population structure in our dataset, we restricted our analyses to include 337,199 White British individuals based on the following five criteria reported by the UK Biobank in the file "ukb_sqc_v2.txt":

1. self-reported white British ancestry ("in_white_British_ancestry_subset" column)
2. used to compute principal components ("used_in_pca_calculation" column)
3. not marked as outliers for heterozygosity and missing rates ("het_missing_outliers" column)
4. do not show putative sex chromosome aneuploidy ("putative_sex_chromosome_aneuploidy" column)
5. have at most 10 putative third-degree relatives ("excess_relatives" column).

We annotated variants using the VEP LOFTEE plugin (https://github.com/konradjk/loftee) and variant quality control by comparing allele frequencies in the UK Biobank and gnomAD version 2.0.1 (gnomad.exomes.r2.0.1.sites.vcf.gz)[12,54].

We focused on variants outside of major histocompatibility complex (MHC) region (chr6:25477797-36448354) based on the Genome Reference Consortium GRCh37 definition[55] and performed LD pruning using PLINK[56] with "--indep 50 5 2". Furthermore, we selected variants according to the following rules:

- Missingness of the variant is less than 1%.
- Minor-allele frequency is greater than 0.01%.
- The variant is in the LD-pruned set.
- Hardy-Weinberg disequilibrium test p-value is greater than $1.0 \times 10^{-7}$.
- Manual cluster plot inspection. We investigated cluster plots for subset of our variants and removed 11 variants that have unreliable genotype calls[12].
- Passed the comparison of minor allele frequency with gnomAD dataset[12,54].

These variant filters are summarized in Supplementary Fig. 1.

**Phenotype data preparation**. We organized 2,138 phenotypes from the UK Biobank[14] in 11 distinct groups (Supplementary Table 1, Supplementary Data 1). We included phenotypes with at least 100 cases for binary phenotypes and 100 individuals with non-missing values for quantitative phenotypes. We used phenotype definitions from a previously published paper[12]. Briefly, a combination of cancer diagnoses from the UK Cancer Register with self-reported diagnoses, a combination of disease diagnoses from the UK National Health Service Hospital Episode Statistics with self-reported diagnoses, and the UK Biobank phenotype category 100034 (Family history–Touchscreen–UK Biobank Assessment Centre) were used for disease outcomes, cancer phenotypes, and family history definitions, respectively[12]. We also used additional data fields and data category from the UK Biobank[14] to define the phenotypes in the following categories as well as 19 and 42 additional miscellaneous binary and quantitative phenotypes: medication, imaging, physical measurements, assays, and binary and quantitative questionnaire Supplementary Table 1, Supplementary Data 1).

Some phenotype information from the UK Biobank contains three instances, each of which corresponds to (1) the initial assessment visit (2006–2010), (2) first repeat assessment visit (2012–2013), and (3) imaging visit (2014–). For binary phenotype, we defined "case" if the participants are classified as case in at least one of their visits and "control" otherwise. For quantitative phenotype, we took a median of non-NA values. In total, we defined 1,196 binary phenotypes and 943 quantitative phenotypes.

**Genome-wide association analyses of 2,138 phenotypes**. Association analyses for single variants were applied to the 2,138 phenotypes separately. We used the following covariates in our GWAS analyses: age, sex, array type, and the first four genetic principal components provided by UK Biobank in the sample QC file[14], where the array type indicates if a participant was genotyped with UK Biobank Axiom Array or UK BiLEVE Axiom Array. For binary phenotypes, we performed Firth-fallback logistic regression using PLINK v2.00a (17 July 2017)[12,56]. For quantitative phenotypes, we applied generalized linear model association analysis with PLINK v2.00a (20 Sep. 2017)[56]. We applied quantile normalization for phenotype (--pheno-quantile-normalize option), where we fit a linear model with covariates and transform the phenotypes to normal distribution $N(0,1)$. while preserving the original rank. We used the following covariates in our analysis: age, sex, types of genotyping array, and the first four genotype principal components computed from the UK Biobank[14].

To test the effects of population stratification correction on the association analysis, we performed additional GWAS with age, sex, types of array, and the first ten genotype principal components as covariates for the five quantitative traits and five binary traits. For each pair of GWAS summary statistics with four and ten genotype principal components, we computed the genetic correlations and confirmed that the two GWAS run yielded the almost identical results (Supplementary Table 5). We used GNU parael in part of our analysis[57].

**Summary statistic matrix construction and variant filters**. We constructed three Z-score summary statistic matrices. Each element of the matrix corresponds to summary statistic for a particular pair of a phenotype and a variant. We imposed different sets of variant filters.

- Variant quality control filter: Our quality control filter described in the previous section on genotype data preparation.
- Non-MHC variant filter: All variants outside of major histocompatibility complex region. With this filter, variants in chr6:25477797-36448354 were excluded from the summary statistic matrix.
- Coding-only: With this filter, we subset to include only the variants having the VEP LOFTEE predicted consequence of: missense, stop gain, frameshift, splice acceptor, splice donor, splice region, loss of start, or loss of stop.
- PTVs-only: With this filter, we subset to include only the variants having the VEP LOFTEE predicted consequence of: stop gain, frameshift, splice acceptor, or splice donor.

By combining these filters, we defined the following sets of variants

- All-non-MHC: This is a combination of our variant QC filter and non-MHC filter.
- Coding-non-MHC: This is a combination of our variant QC filter, non-MHC filter, and Coding-only filter.
- PTVs-non-MHC: This is a combination of our variant QC filter, non-MHC filter, and PTVs-only filter.

In addition to phenotype quality control and variant filters, we introduced value-based filters based on statistical significance to construct summary statistic matrices only with confident values. We applied the following criteria for the value filter:

- P-value of marginal association is less than 0.001.
- Standard error of beta value or log odds ratio is less than 0.08 for quantitative phenotypes and 0.2 for binary phenotypes.

With these filters, we obtained the following two matrices:

- "All-non-MHC" dataset that contains 2,138 phenotypes and 235,907 variants. We label this dataset as **"all" dataset**.
- "Coding-non-MHC" dataset that contains 2,064 phenotypes and 16,135 variants. We label this dataset as **"Coding only" dataset**.
- "PTVs-non-MHC" dataset that contains 628 phenotypes and 784 variants. We label this dataset as **"PTVs only" dataset**.

The coding-only and PTVs-only datasets contain a fewer number of phenotypes because not all the phenotypes have statistically significant associations with coding variants or PTVs. The effects of variant filters are summarized in Supplementary Fig. 1. Finally, we transformed the summary statistics to Z-scores so that each vector that corresponds to a particular phenotype has zero mean with unit variance.

**TSVD of the summary statistic matrix.** For each summary statistic matrix, we applied truncated singular value decomposition (TSVD). The matrix, which we denote as $\mathbf{W}$, of size $N \times M$, where $N$ denotes the number of phenotypes and $M$ denotes the number of variants, is the input data. With TSVD, $\mathbf{W}$ is factorized into a product of three matrices: $\mathbf{U}$, $\mathbf{S}$, and $\mathbf{V}^T$: $\mathbf{W} = \mathbf{USV}^T$, where $\mathbf{U} = (u_{i,k})_{i,k}$ is an orthonormal matrix of size $N \times K$ whose columns are phenotype (left) singular vectors, $\mathbf{S}$ is a diagonal matrix of size $K \times K$ whose elements are singular values, and $\mathbf{V} = (v_{j,k})_{j,k}$ is an orthonormal matrix of size $M \times K$ whose columns are variant (right) singular vectors. While singular values in $\mathbf{S}$ represent the magnitude of the components, singular vectors in $\mathbf{U}$ and $\mathbf{V}$ summarize the strength of association between phenotype and component and variant and component, respectively. With this decomposition, the $k$-th latent component (principal component, PC $k$) are represented as a produ of $k$-th column $\mathbf{U}$, $k$-th diagonal element in $\mathbf{S}$, and $k$-th row of $\mathbf{V}^T$. For TSVD on the summary statistics, we used implicitly restarted Lanczos bidiagonalization algorithm (IRLBA)[58] (https://github.com/bw.ewis/lba) implemented on SciDB[59] to compute the first $K$ components in this decomposition.

**Relative variance explained by each of the components.** A scree plot (Supplementary Fig. 2) quantify the variance explained by each component: variance explained by $k$-th component = $s_k^2/\text{Var}_{\text{Tot}}(\mathbf{W})$ where, $s_k$ is the $k$-th diagonal element in the diagonal matrix $\mathbf{S}$ and $\text{Var}_{\text{Tot}}(\mathbf{W})$ is the total variance of the original matrix before DeGAs is applied.

**Selection of number of latent components in TSVD.** In order to apply TSVD to the input matrix, the number of components should be specified. We apply $K = 100$ for our analysis for all of the datasets. Following a standard practice of keeping components with eigenvalues greater than the average[24], we first computed the expected value of squared eigenvalues under the null model where the distribution of variance explained scores across the full-ranks are uniform. This can be computed with the rank of the original matrix, which is equal to the number of phenotypes in our datasets:

$$E[\text{Variance explained by } k-\text{th component under the null}]$$
$$= 1/\left(\text{Rank}(\mathbf{W})^2\right) \tag{1}$$

We then compared the eigenvalues characterized from TSVD with the expected value. For all of the three datasets, we found that that of 100-th component is greater than the expectation. This indicates even the 100-th components are informative to represent the variance of the original matrix. In the interest of computational efficiency, we set $K = 100$.

To demonstrate the robustness of the DeGAs components with respect to the number of latent components ($K$), we performed additional analyses with $K = 90$ and $K = 110$, and investigated the first five latent components as well as the top three components for the three phenotypes of our interest.

**Factor scores.** From these decomposed matrices, we computed **factor score** matrices for both phenotypes and variants as the product of singular vector matrix and singular values. We denote the one for phenotypes as $\mathbf{F}_p = \left(f^p_{i,j}\right)_{i,j}$ the one for variants as $\mathbf{F}_v = \left(f^v_{i,j}\right)_{i,j}$ and defined as follows:

$$\mathbf{F}_p = \mathbf{US} \tag{2}$$

$$\mathbf{F}_v = \mathbf{VS} \tag{3}$$

Since these factor scores are mathematically the same as principal components in principal component analysis (PCA), one can investigate the contribution of the phenotypes or variants for specific principal components by simply plotting factor scores[24] (Fig. 2a, b). Specifically, phenotype factor score is the same as phenotype principal components and variant factor score is the same as variant principal components. By normalizing these factor scores, one can compute contribution scores and cosine scores to quantify the importance of phenotypes, variants, and principal components as described below.

**Scatter plot visualization with biplot annotations.** To investigate the relationship between phenotype and variants in the TSVD eigenspace, we used a variant of biplot visualization[60,61]. Specifically, we display phenotypes projected on

phenotype principal components ($\mathbf{F}_p = \mathbf{US}$) as a scatter plot. We also show variants projected on variant principal components ($\mathbf{F}_v = \mathbf{VS}$) as a separate scatter plot and added phenotype singular vectors ($\mathbf{U}$) as arrows on the plot using sub-axes (Figs. 2b, 4c, Supplementary Figs. 10–11). In scatter plot with biplot annotation, the inner product of a genetic variant and a phenotype represents the direction and the strength of the projection of the genetic association of the variant-phenotype pair on the displayed latent components. For example, when a variant and a phenotype share the same direction on the annotated scatter plot, that means the projection of the genetic associations of the variant-phenotype pair on the displayed latent components is positive. When a variant-phenotype pair is projected on the same line, but on the opposite direction, the projection of the genetic associations on the shown latent components is negative. When the variant and phenotype vectors are orthogonal or one of the vectors are of zero length, the projection of the genetic associations of the variant-phenotype pair on the displayed latent components is zero. Given the high dimensionality of the input summary statistic matrix, we selected relevant phenotypes to display to help the interpretation of genetic associations in the context of these traits.

**Contribution scores.** To quantify the contribution of the phenotypes, variants, and genes to a given component, we computed contribution scores. We first defined phenotype contribution score and variant contribution score. We denote phenotype contribution score and variant contribution score for some component $k$ as $\text{cntr}^{\text{phe}}_k(i)$ and $\text{cntr}^{\text{var}}_k(j)$, respectively. They were defined by squaring the left and right singular vectors and normalizing them by Euclidean norm across phenotypes and variants:

$$\text{cntr}^{\text{phe}}_k(i) = \left(u_{i,k}\right)^2 \tag{4}$$

$$\text{cntr}^{\text{var}}_k(j) = \left(v_{i,k}\right)^2 \tag{5}$$

where, $i$ and $j$ denote indices for phenotype and variant, respectively. Because $U$ and $V$ are orthonormal, the sum of phenotype and variant contribution scores for a given component are guaranteed to be one, i.e. $\sum_i \text{cntr}^{\text{phe}}_k(i) = \sum_j \text{cntr}^{\text{var}}_k(j) = 1$.

Based on the variant contribution scores for the $k$-th component, we defined the gene contribution score for some component $k$ as the sum of variant contribution scores for the set of variants in the gene:

$$\text{cntr}^{\text{gene}}_k(g) = \sum_{j \in g} \text{cntr}^{\text{var}}_k(j) \tag{6}$$

where, $g$ denotes indices for the set of variants in gene $g$. To guarantee that gene contribution scores for a given component sum up to one, we treated the variant contribution score for the non-coding variants as gene contribution score. When multiple genes, $g_1, g_2, \ldots, g_n$ are sharing the same variants, we defined the gene contribution score for the intersection of multiple genes rather than each gene:

$$\text{cntr}^{\text{gene}}_k(\{g_i | i \in [1, n]\}) = \sum_{\{j | j \in g_1 \wedge j \in g_2 \wedge \cdots \wedge j \in g_n\}} \text{cntr}^{\text{var}}_k(j) \tag{7}$$

With these contribution scores for a given component, it is possible to quantify the relative importance of a phenotype, variant, or gene to the component. Since DeGAs identifies latent components using unsupervised learning, we interpret each component in terms of the driving phenotypes, variants, and genes, i.e. the ones with large contribution scores for the component.

The top 20 driving phenotypes, variants, and genes (based on contribution scores) for the top five TSVD components and the top three key components for our phenotypes of interest are summarized in Supplementary Table 2.

We used stacked bar plots for visualization of the contribution score profile for each of the components. We represent phenotypes, genes, or variants with large contribution scores as colored segments and aggregated contributions from the remaining ones as "others" in the plot (Figs. 2c, d, 3a, 4a, b, Supplementary Fig. 4). To help interpretation of the major contributing factors for the key components, we grouped phenotypes into categories, such as "fat", "fat-free" phenotypes, and showed the sum of contribution scores for the phenotype groups. The list of phenotype groups used in the visualization is summarized in Supplementary Table 2.

**Squared cosine scores.** Conversely, we can also quantify the relative importance of the latent components for a given phenotype or variant with squared cosine scores. We denote phenotype squared cosine score for a given phenotype $i$ and variant squared cosine score for a given variant $j$ as $\cos^{2^{\text{phe}}}_i(k)$ and $\cos^{2^{\text{var}}}_j(k)$, respectively. They are defined by squaring of the factor scores and normalizing them by Euclidean norm across components:

$$\cos^{2^{\text{phe}}}_i(k) = \left(f^p_{i,k}\right)^2 / \sum_{k'} \left(f^p_{i,k'}\right)^2 \tag{8}$$

$$\cos^{2^{\text{var}}}_j(k) = \left(f^v_{j,k}\right)^2 / \sum_{k'} \left(f^v_{j,k'}\right)^2 \tag{9}$$

By definition, the sum of squared cosine scores across a latent component for a given phenotype or variant equals to one, i.e. $\sum_k \cos^{2^{\text{phe}}}_i(k) = \sum_k \cos^{2^{\text{var}}}_j(k)$. While singular values in the diagonal matrix $S$ quantify the importance of latent components for the global latent structure, the phenotype or variant squared cosine

score quantifies the relative importance of each component in the context of a given phenotype or a variant. The squared cosine scores for the phenotypes highlighted in the study is summarized in Supplementary Figs. 3 and 9.

Note that squared cosine scores and contribution scores are two complementary scoring metrics to quantify the relationship among phenotypes, components, variants, and genes. It does not necessarily have the inverse mapping property. For example, it is possible to see a situation, where for a given phenotype $p$, phenotype squared cosine score identifies $k$ as the top key component, but phenotype contribution score for $k$ identifies $p'$ ($p' \neq p$) as the top driving phenotype for the component $k$. This is because the two scores, contribution score and squared cosine score, are both defined by normalizing singular vector and principal component vector matrices, respectively, but with respect to different slices: one for row and the other for column.

**TSVD of the individual-level phenotypes**. To characterize the latent components in the raw phenotype data, we first applied median imputation for missing values on the phenotype data followed by Z-score transformation. Using Python scikit-learn package[62], we applied TSVD on the imputed and normalized phenotype matrix and characterized the first five latent components and visualized the scree plot as well as the phenotype and individual PCs in scatter plots.

**Genome-wide-association analysis for phenotype PCs**. Using the results of the phenotype decomposition described above, we defined principal components of the individual's phenotype (phenotype PCs) and applied genome-wide association analysis using the same procedure we used for the original quantitative traits. We used R package qqplot to generate Manhattan plot[63].

**Genetic correlation of phenotype PCs**. To compare the results of association analysis of phenotype PCs, we computed genetic correlation using LD score regression[64]. We summarized the estimated genetic correlation ($r_g$) as heatmap and characterized the median value of absolute value of $r_g$ among the top $k$ phenotype PCs as a function of $k$.

**Genomic region enrichment analysis with GREAT**. We applied the genomic region enrichment analysis tool (GREAT version 4.0.3) to each DeGAs components[20,21]. We used the mouse genome informatics (MGI) phenotype ontology, which contains manually curated knowledge about hierarchical structure of phenotypes and genotype-phenotype mapping of mouse[33]. We downloaded their ontologies on 2017-09-28 and mapped MGI gene identifiers to Ensembl human gene ID through unambiguous one-to-one homology mapping between human and mouse Ensembl IDs. We removed ontology terms that were labeled as "obsolete", "bad", or "unknown" from our analysis. As a result, we obtained 709,451 mapping annotation spanning between 9,592 human genes and 9,554 mouse phenotypes.

For each DeGAs component, we selected the top 5,000 variants according to their variant contribution score and performed enrichment analysis with the default parameters[20,21]. Since we included the non-coding variants in the analysis, we focused on GREAT binomial genomic region enrichment analysis based on the size of regulatory domain of genes and quantified the significance of enrichment in terms of binomial fold enrichment and binomial $p$-value. Given that we have 9,554 terms in the ontology, we set a Bonferroni-corrected $p$-value threshold of $5 \times 10^{-6}$.

To illustrate the results of the genomic region enrichment analysis for the phenotypes of our interest, we made circular bar plots using the R package ggplot2[65], where each of the key components are displayed in the innermost track with their phenotype squared cosine score to be proportional to their angle, and the resulted significant ontology terms are represented as the bars. To focus on the significant signals with large effect size, we imposed additional filter of binomial fold $\geq 2.0$ and binomial $p$-value threshold of $5 \times 10^{-7}$, The binomial fold change is represented as the radius and the binomial $p$-value is represented as color gradient in a log scale in the plot (Fig. 3b, Supplementary Figs. 7-8, Supplementary Data 3–5).

**Specificity analysis of GREAT enrichment**. To test the specificity of the GREAT enrichment of each of the 100 DeGAs components, we computed Jaccard index similarity scores. For each DeGAs latent component, we looked at the GREAT enrichment and took the top five enriched terms sorted by GREAT binomial fold. To measure the similarity between these enriched terms, we identified the set of genes annotated for those terms and computed Jaccard index defined below:

$$\text{Similarity}(\text{term set}_A, \text{term set}_B)$$
$$= \frac{|\text{Gene set}(\text{term set}_A) \cap \text{Gene set}(\text{term set}_B)|}{|\text{Gene set}(\text{term set}_A) \cup \text{Gene set}(\text{term set}_B)|} \quad (10)$$

where,

$$\text{Gene set}(\text{term set}_A) = \bigcup_{t \in A} \text{Gene set}(\text{term}_t) \quad (11)$$

and Gene set($\text{term}_t$) indicates set of genes annotated with term $t$. We computed all

the pair-wise similarity across the top $k$ DeGAs components and summarized their median as a function of $k$.

**Quality control of variant calling with intensity plots**. To investigate the quality of variant calling for the two PTVs highlighted in the study, we manually inspected intensity plots. These plots are available on Global Biobank Engine.

https://biobankengine.stanford.edu/intensity/rs114285050
https://biobankengine.stanford.edu/intensity/rs150090666

**Phenome-wide association analysis**. To explore the functional roles of the two PTVs across thousands of potentially correlated phenotypes, we performed a phenome-wide association study (PheWAS). We report the statistically significant ($p < 0.001$) associations with phenotypes with at least 1000 case count or 1000 individuals with measurements with non-missing values for binary and quantitative phenotypes, respectively (Fig. 3d, Supplementary Fig. 10). The results of this PheWAS are also available as interactive plots as a part of Global Biobank Engine.

https://biobankengine.stanford.edu/variant/5-145895394
https://biobankengine.stanford.edu/variant/11-14865399

**Univariate regression analysis for the identified PTVs**. To quantify the effects of the two PTVs on obesity, we performed univariate regression analysis. We extracted individual-level genotype information for the two PTVs with the PLINK2 pgen Python API (http://www.cog-genomics.org/plink/2.0/)[56]. After removing individuals with missing values for BMI and genotype, we performed linear regression for BMI (http://biobank.ctsu.ox.ac.uk/crystal/field.cgi?id=21001) with age, sex, and the first four genomic PCs as covariates:

$$\text{BMI} \sim 0 + \text{age} + \text{as.factor(sex)} + \text{PC1} + \text{PC2} + \text{PC3}$$
$$+ \text{PC4} + \text{as.factor(PTV)} \quad (12)$$

where, PC1-4 denotes the first four genomic principal components, PTV ranges in 0, 1, or 2 and it indicates the number of minor alleles that the individuals have.

**The polygenicity analysis and phenotype contribution scores**. To quantify the polygenicity of phenotypes, we applied LD clumping procedure to select and count independent set of SNPs associated with the trait using PLINK v1.90b6.7 (2 December 2018)[56] with --clump --clump-p1 1e-4. We compared the number of independent associations (clumped "hits") with the DeGAs contribution scores and summarized as scatter plots (Supplementary Fig. 29).

**Mouse 3T3-L1 cell culture and differentiation**. 3T3-L1 preadipocytes were obtained from ATCC (Catalogue # CL-173™) and cultured in Dulbecco's Modified Eagle's Medium (DMEM) containing 10% fetal bovine serum (FBS) and antibiotics (100 U mL$^{-1}$ of penicillin G and 100 μg mL$^{-1}$ of streptomycin) at 37 °C in a humidified atmosphere containing 5% $CO_2$. To obtain fully differentiated adipocytes, 3T3-L1 preadipocytes were grown into 2-day post-confluence, and then differentiation was induced by using a standard differentiation cocktail containing 0.5 mM of IBMX, 1 μM of dexamethasone, 1 μg mL$^{-1}$ of insulin, and 10% FBS. After 48 h, medium was changed into DMEM supplemented with 10% FBS and 1 μg mL$^{-1}$ of insulin and replenished every 48 h for an additional 6 days.

**Human SGBS cell culture and differentiation**. SGBS cells, a gift from Dr. Martin Wabitsch (Univ. of Ulm, Germany), were cultured in DMEM/F12 containing 33 μM biotin, 17 μM pantothenate, 0.1 mg mg$^{-1}$ streptomycin and 100 U mL$^{-1}$ penicillin (0F medium) supplemented with 10% FBS in a 5% $CO_2$ incubator. To initiate differentiation, confluent cells were stimulated by 0F media supplemented with 0.01 mg mL$^{-1}$ human transferrin, 0.2 nM T3, 100 nM cortisol, 20 nM insulin, 250 μM IBMX, 25 nM dexamethasone and 2 μM rosiglitazone. After day 4, the differentiating cells were kept in 0F media supplemented with 0.01 mg mL$^{-1}$ human transferrin, 100 nM cortisol, 20 nM insulin and 0.2 nM T3 for additional 8–10 days until cells were fully differentiated.

**siRNA knockdown in 3T3-L1 preadipocytes**. At 80% confluence, 3T3-L1 pre-adipocytes were transfected with 50 nM siRNA against *Gpr151* (Origene #SR413989, 3 unique siRNA duplexes), *Pde3b* (Origene #SR422062, 3 unique siRNA duplexes), or scrambled negative control (Origene #SR30004) using Lipofectamine™ RNAiMAX Transfection Reagent (Invitrogen) following the manufacturer's protocol. The 3 siRNAs against the same target were diluted in a 1:1:1 ratio to reach the total concentration of 50 nM, when transfecting as a pool. The transfected cells were incubated for 48 h and then subjected to differentiation.

**Reverse transcription (RT) and qPCR analysis**. Total RNA was extracted using TRIzol reagent (Invitrogen), following the manufacturer's instruction. RNA was converted to cDNA using High-Capacity cDNA Reverse Transcription Kit (Applied Biosystems). Quantitative PCR reactions were prepared with TaqMan™

Fast Advanced Master Mix (Thermo Fisher Scientific) and performed on ViiA 7 Real-Time PCR System (Thermo Fisher Scientific). All data were normalized to the content of Cyclophilin A (PPIA), as the endogenous control. TaqMan primer information for RT-qPCR is listed below: *GPR151* (Hs00972208_s1), *Gpr151* (Mm00808987_s1), *PDE3B* (Hs00265322_m1), *Pde3b* (Mm00691635_m1), *Pparg* (Mm00440940_m1), *Cebpa* (Mm00514283_s1), *Fabp4* (Mm00445878_m1), *PPIA* (Hs04194521_s1), *Ppia* (Mm02342430_g1).

**Oil Red O staining and quantification**. Cells were washed twice with PBS and fixed with 10% formalin for 1 h at room temperature. Cells were then washed with 60% isopropanol and stained for 15 min with a filtered Oil Red O solution (mix six parts of 0.35% Oil Red O in isopropanol with four parts of water). After washing with PBS 4 times, cells were maintained in PBS and visualized by inverted microscope. After taking pictures, Oil Red O stain was extracted with 100% isopropanol and the absorbance was measured at 492 nm by a multi-well spectrophotometer (Bio-Rad).

**Lipolysis assay**. Glycerol release into the culture medium was used as an index of lipolysis. Fully differentiated 3T3-L1 adipocytes were serum starved overnight and then treated with either vehicle (DMSO) or the lipolytic stimuli isoproterenol (ISO, 10 μM) for 3 h. The culture medium was collected and the glycerol content in the culture medium was measured using an adipocyte lipolysis assay kit (ZenBio #LIP-1-NCL1). Glycerol release into the culture medium was normalized to the protein content of the cells from the same plate.

**Overexpression of *GPR151* in 3T3-L1 preadipocytes**. The *GPR151* construct was obtained from Addgene (#66327). This construct includes a cleavable HA signal to promote membrane localization, a FLAG epitope sequence for cell surface staining followed by codon-optimized human *GPR151* sequence[66]. We PCR-amplified the above sequence with stop codon and assembled it into a lentiviral plasmid (Addgene #85969) with either EF1α promoter (Addgene #11154) or aP2 promoter (Addgene #11424). EF1α-*GPR151* or aP2-*GPR151* lentiviral plasmid were transfected into human embryonic kidney 293T cells, together with the viral packaging vectors pCMV-dR8.91 and pMD2-G. 72 h after transfection, virus-containing medium was collected, filtered through a 0.45-μm pore-size syringe filter, and frozen at −80 °C. 3T3-L1 preadipocytes at 50% confluence were infected with the lentivirus stocks containing 8 μg mL$^{-1}$ polybrene. Two days after transduction, lentivirus-infected 3T3-L1 preadipocytes were subject to differentiation.

**Flow cytometry analysis**. Day 6 differentiating 3T3-L1 adipocytes were collected and washed with ice cold FACS buffer (PBS containing 2% BSA). Cells were first resuspended into FACS staining buffer (BioLegend # 420201) at ~1 M cells per 100 μl and incubated with anti-mouse CD16/CD32 Fc Block (BioLegend #101319) at room temperature for 10–15 min. Cells were then incubated with APC-conjugated FLAG antibody (BioLegend #637307) for 20–30 min at room temperature in the dark. Following washing and centrifugation, cells were resuspended in FACS buffer and sorted using a BD Influx™ Cell Sorter. Cells without FLAG antibody staining were used to determine background fluorescence levels. Cells were sorted based on APC fluorescence and collected directly into TRIzol reagent for RNA extraction.

**Western blot analysis**. Lysate aliquots containing 50 μg of proteins were denatured, separated on a 4–10% SDS-polyacrylamide gel, and transferred to nitrocellulose membranes using a Trans-Blot® SD Semi-Dry Transfer Cell (Bio-Rad). Membranes were blocked in 5% non-fat milk and incubated overnight at 4 °C with primary antibodies: anti-GPR151 (LSBio # LS-B6760-50, 1:500 dilution) or anti-beta-actin (Cell Signaling #3700, 1:1000 dilution). Subsequently, the membranes were incubated for 1 h at room temperature with IRDye® 800CW goat-anti-mouse antibody (LI-COR #926-32210, 1:2500 dilution). Target proteins were visualized using Odyssey® Fc Imaging System (LI-COR). Uncropped blots were provided in the Source Data.

**Statistical analysis of functional data**. Data are expressed as mean ± SEM. Student's *t*-test was used for single variables, and one-way ANOVA with Bonferroni post hoc correction was used for multiple comparisons using GraphPad Prism 7 software.

**Reporting summary**. Further information on research design is available in the Nature Research Reporting Summary linked to this article.

## Data availability

The association analysis data, the interactive DeGAs App, and its video tutorial are available as a part of Global Biobank Engine (https://biobankengine.stanford.edu/degas). The decomposed matrices are available on figshare (https://doi.org/10.35092/yhjc.9202247). The source data underlying Figs. 2a, b, 4c, and 5 and Supplementary Figs. 27, 28 are provided as a Source Data file.

## Code availability

Analysis scripts and notebooks are available on GitHub (https://github.com/rivas-lab/public-resources/tree/master/uk_biobank/DeGAs).

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

## Acknowledgements

This research has been conducted using the UK Biobank Resource under Application Number 24983, "Generating effective therapeutic hypotheses from genomic and hospital linkage data" (http://www.ukbiobank.ac.uk/wp-content/uploads/2017/06/24983-Dr-Manuel-Rivas.pdf). We thank all the participants in the UK Biobank study. We thank Robert Tibshirani, Nasa Sinnott-Armstrong, the members of the Rivas lab, the Ingelsson lab, and the Bejerano lab for helpful feedback. This work was supported by National Human Genome Research Institute (NHGRI) and National Institute of Diabetes and Digestive and Kidney Diseases (NIDDK) of the National Institutes of Health (NIH) under awards R01HG010140 and R01DK106236. The content is solely the responsibility of the authors and does not necessarily represent the official views of the National Institutes of Health. Y.T. is supported by a Funai Overseas Scholarship from the Funai Foundation for Information Technology and the Stanford University School of Medicine. J.M.J. was funded by grant NNF17OC0025806 from the Novo Nordisk Foundation and the Stanford Bio-X Program. M.A.R. and C.D. are supported by Stanford University and a National Institute of Health center for Multi- and Trans-ethnic Mapping of Mendelian and Complex Diseases grant (5U01 HG009080). C.D. is supported by a postdoctoral fellowship from the Stanford Center for Computational, Evolutionary, and Human Genomics and the Stanford ChEM-H Institute. We would like to thank the Customer Solutions Team from Paradigm4 who helped us implement efficient databases for queries and application of inference methods to the data, and also implemented optimized versions of truncated singular value decomposition.

## Author contributions

M.A.R. and E.I. conceived and designed the study. Y.T. and M.A.R. carried out the statistical and computational analyses with advice from J.M.J., H.H., M.A., C.D., B.N., K.L, T.H., G.B. and E.I. J.L., C.Y.P. and E.I. carried out the functional experiments. Y.T., M.A. and C.D. carried out quality control of the data. C.C. optimized and implemented computational methods. Y.T. and M.A.R. developed the DeGAs app in Global Biobank Engine. M.A.R. supervised computational and statistical aspects of the study. E.I. supervised experimental aspects of the study. The manuscript was written by Y.T., J.L., J.M.J., E.I. and M.A.R; and revised by all the co-authors. All co-authors have approved of the final version of the manuscript.

## Additional information

**Competing interests:** The authors declare no competing interests.

