## [Peer Review File · Nature Communications]

Editorial Note: This manuscript has been previously reviewed at another journal that is not operating a transparent peer review scheme. This document only contains reviewer comments and rebuttal letters for versions considered at Nature Communications .

Reviewers' comments:

Reviewer #1 (Remarks to the Author):

The authors have done a very good job in answering the questions.

Reviewer #2 (Remarks to the Author):

The authors write a short rebuttal to the reviewers and address some of the comments; however, some of the responses provide little quantitative experimentation to address the comments.

Specifically:

In response to reviewer 1, they write in point (4):

"As we summarized in the results and discussion sections in our manuscript, we don't see a clear separation of relevant phenotypes, variants, and genes with this alternative approach..."

- why did the alternative approach not work? The reason for the contrast is not given and this of major importance since many more individual-level data will be released in the near future.

In response to reviewer 2, they write in point (12):

"If one takes GWAS summary statistics from a different GWAS study, the latent components can be different due to the difference in experimental design, such as selection of phenotypes and variants in the association analysis. "

- This is speculative. Can the authors test these hypotheses in the current data? This is related to the point (4) above.

In point (14), the authors write:

"Contribution scores quantify relative importance of a phenotype, variant, or gene to a given component and is defined based on the squared distance of a phenotype, variant, or gene from the origin (Fig. 1d, Methods). Using scores, DeGAs identifies the key latent components for a given complex trait and annotated them with the driving phenotypes, genes, and variants (Fig. 1c, Methods). We performed biological characterization of DeGAs components with the genomic region enrichment analysis tool (GREAT) followed by functional experiments in adipocytes (Fig. 1e)."

- The authors are still not describing concretely how to take the GREAT annotations and the scores to rank order the latent components. As a reader and someone who wants to apply the method, I have no idea on how to do so.

In point (15), the authors write:

"Thank you for pointing out the interesting aspects of our results. Anthropometric traits are known to be highly polygenic (Locke et al. Nature, 2015, Yengo et al. bioRxiv, 2018) and we think that is one way to interpret the results. Also, it is possible that anthropometric traits are dominating among the continuous traits with full sample size, since we didn't include biomarkers dataset in our analysis. "

- Again, why not test these speculative comments? First, many traits — not only anthropomorphic - are known to be highly polygenic and this cannot explain the results fully. If the authors think it is a sample size issue, why not test?

Reviewer #3 (Remarks to the Author):

In their manuscript Tanigawa and colleagues describe a broad genomic screen of more than 2100 phenotypes in about 340,000 subjects. The authors applied a truncated singular value decomposition (DeGAs) with a particular focus on the traits B body mass index (BMI), myocardial infarction (MI) and gallstones, finally focusing on two putative loss of function genes, GPR115 and PDE3b and their role on adipocyte function.

The first part of the paper is interesting and innovative because it deals with a huge number of human data and provides bioinformatics strategies for the identification of key components that might be responsible for complex phenotypes. The results of complex analysis which also included information on mouse phenotype data are lists of genes that appear to play a role as risk factors, for the disease and / or comorbidities. However, unfortunately the success of the approach is not proven in the last part of the manuscript. It is also not clear why the authors selected GPR115 and PDE3B for the in vitro analysis and not LIPT1 and MLPH, or two candidates from two different PC.

The data on the characterization of both genes are not convincing. GPR115 exhibits a very low expression in adipose tissue and in 3T3-L1 and SGBS cells (with not changes during differentiation). Therefore, it is difficult to understand the effect on reduced expression of adipogenic genes like PPARG. The question is if the knockdown is indeed specific and if a second siRNA has been tested and if off-target effects can be excluded. According to the literature, GPR115 is supposed to be involved in skin development. However, the knockout mouse generated and analyzed by the IMPC shows – similar to the experiments in cells a reduction of total fat mass. However, a large number of ko mice have an impaired fat accumulation which might have several reasons, often mediated via secondary effects. In addition, the suppression of PDE3B was without effect, the IMPC ko shows no effect on body weight or fat mass but a bone phenotype (long tibia).

I recommend to either removing the data of the two candidates or at least to shorten it strongly and to significantly reduce the interpretation. The classification of both candidates as putative novel therapeutic targets is truly an overstatement.

Annette Schürmann

We thank the reviewers for their constructive comments and their time. We believe that the changes made in the light of their comments have significantly improved the manuscript.

Our responses to the reviewers below are in blue font, the comments from the reviewer are copied in black, and quoted texts from the updated manuscript are shown in gray with a vertical bar (examples are shown below):

Response to our Reviewer 1

The authors have done a very good job in answering the questions.

Thank you very much for your time and valuable feedback on our manuscript.

Response to our Reviewer 2

(19) Comparison with an alternative individual-level approach

In response to reviewer 1, they write in point (4):

“As we summarized in the results and discussion sections in our manuscript, we don’t see a clear separation of relevant phenotypes, variants, and genes with this alternative approach...”
- why did the alternative approach not work? The reason for the contrast is not given and this of major importance since many more individual-level data will be released in the near future.

Thank you for giving us the opportunity to clarify the objectives of the additional analysis. The original question from reviewer 1 was a suggestion of an alternative approach: to utilize the individual-level data on the phenotypes to characterize the latent structures and to study genetic associations using those phenotypic latent structures.

We followed the suggestion from reviewer 1 and performed the following analysis as outlined in the previous response: (i) latent structure characterization of phenotype data (Supplementary Figures S4-S5); (ii) GWAS (Supplementary Figure S6); and (iii) genetic correlation analysis among the derived summary statistics (Supplementary Figures S7-S8).

In this analysis, we found that the latent structures characterized from the individual-level phenotype data picked up phenotypes like “traffic intensity on the major roads,” which does not have a clear genetic component (Supplementary Figures S5, quoted below).

This observation is not surprising when one considers a property of the TSVD algorithm: it finds the latent components that explains most of the variation in the input dataset. When we apply TSVD on the individual-level phenotype data as suggested in the alternative approach, it identifies (1) latent components that explains most of the phenotypic variance, and (2) sets of phenotypes that show most of the correlation with the identified phenotypic latent components. Specifically, there is no guarantee that the identified components represent latent structures in the genetic associations. The captured phenotypic variation may come from different sources, such as environmental factors and technical and/or non-technical errors in the phenotypic measurements. In contrast, our proposed method, DeGAs, directly operates on the genetic association data (represented as a summary statistic matrix) and captures most of the variation in genetic associations in the identified DeGAs components.

While pursuing the additional quantitative analysis for the suggested alternative approach, we also observed that the genetic associations characterized for the phenotypic latent components were not independent when we computed the genetic correlation among them (Supplementary Figures S7-S8). This highlights the pleiotropic effects of genetic variants even when the phenotypes are mathematically orthogonal among others. In contrast, the latent components

from our proposed approach, DeGAs, are guaranteed to be orthogonal among other given the way we decomposed the summary statistic matrix.

As you pointed out, the UK Biobank will release more individual-level data enabling us to access more phenotypes and/or reduce the number of missing individuals for the existing phenotypes. However, even with the new individual-level data, it is unclear whether the alternative approach will identify latent components of genetic associations because of the aforementioned properties of the alternative approach. With more individual-level data, our DeGAs approach will also benefit because that will enable us to obtain more accurate estimate of genetic associations (the standard errors from GWAS will decrease as the number of samples with non-missing phenotypic values increases).

Together, the suggested alternative approach provided a different perspective and interpretation, and the comparison highlights that DeGAs approach can focus on the variations in the genome-wide associations (not the phenotypic variations which may include environmental factors) across >2,000 traits and identify their orthogonal latent components. The results of those additional analyses and their implications are summarized in the Results and Discussion sections in the main texts (quoted below).

Results (pp. 3-4, line 131-140):

To highlight the ability of DeGAs to capture related sets of phenotypes, genes, and variants in genetic associations, we also applied TSVD to the missing-value imputed and Z-score transformed phenotype matrix and characterized the first 100 latent components (Methods). Using the individual and phenotype PCA plots, we found a fewer number of components that explains most of the variance and several phenotypes, such as traffic intensity of the nearest major road and creatinine (enzymatic) in urine, are dominantly driving the top phenotypic PCs (Supplementary Fig. S4-S5). We applied GWAS for each of the decomposed phenotypes (Supplementary Fig. S6). Through the genetic correlation analysis with the derived summary statistics, we found non-zero genetic correlations among the phenotypic PCs (Supplementary Fig. S7-S8).

Discussion (p. 8, line 318-323):

Our comparison of DeGAs to an alternative approach – decomposition of individual phenotype data followed by GWAS – highlights the ability of DeGAs to capture most of the variation in the genetic associations and to enable identification of biomedically relevant genetic signals as latent components connecting sets of genetic variants and phenotypes.

Thank you very much for your question. It helped us to clarify the properties of DeGAs in a comprehensive manner.

(20) Latent components from different GWAS study

In response to reviewer 2 (12):

“If one takes GWAS summary statistics from a different GWAS study, the latent components can be different due to the difference in experimental design, such as selection of phenotypes and variants in the association analysis. “

- This is speculative. Can the authors test these hypotheses in the current data? This is related to the point (4) above.

In our manuscript, we analyzed three datasets: “all” variants with both coding and non-coding variants, coding variants, and protein-truncating variants. These datasets contain different sets of variants and phenotypes (p.2, line 85-86):

we performed separate analyses on three variant sets: (1) all directly-genotyped variants, (2) coding variants, and (3) PTVs (Supplementary Fig. S1).

and (p.3, line 90-93):

N and M denote the number of phenotypes and variants, respectively. N and M were 2,138 and 235,907 for the “all” variant group; 2,064 and 16,135 for the “coding” variant group; and 628 and 784 for the PTV group.

For each of the three datasets, we quantified the contributions of phenotypes and genetic variants and found that they are different (Figures 3a, 4a, 4b, Supplementary Figures S9, S21, and S22). For example, the top contributing genes for the most important component for BMI were *FTO*, *MC4R*, and *PDE3B*, respectively for “all” variant group, “coding” variant group, and PTV group, respectively as shown below:

We clarified this property in the discussion section (p.8, line 339-344):

Due to differences in phenotype and variant selection, it is possible that the latent structures discovered from DeGAs would be different if using GWAS summary statistics from a different GWAS study. However, DeGAs is capable of identifying the most relevant components for a given input dataset using quantitative scores. In fact, our analysis for the three datasets – “all”, coding, and PTVs – identified different PCs for each trait of our interest, but their characterization with contribution scores enabled interpretation of the DeGAs components.

(21) application of GREAT enrichment analysis for biological characterization of DeGAs components

In point (14), the authors write:

“Contribution scores quantify relative importance of a phenotype, variant, or gene to a given component and is defined based on the squared distance of a phenotype, variant, or gene from the origin (Fig. 1d, Methods). Using scores, DeGAs identifies the key latent components for a given complex trait and annotated them with the driving phenotypes, genes, and variants (Fig. 1c, Methods). We performed biological characterization of DeGAs components with the genomic region enrichment analysis tool (GREAT) followed by functional experiments in adipocytes (Fig. 1e).”

- The authors are still not describing concretely how to take the GREAT annotations and the scores to rank order the latent components. As a reader and someone who wants to apply the method, I have no idea on how to do so.

For readers that want to apply our methods, we provide an interactive web application as a part of the Global Biobank Engine so that people in the community can browse the results and use them for their analysis. On the DeGAs app page, we have prepared a video tutorial for our app so that the new users can easily learn our method and use it to address their research questions. We also provide the analysis code available on GitHub. We clarified those points in the data and code availability section quoted below (p. 21, lines 834-838):

Data and code availability:

The association analysis data, the interactive DeGAs App, and its video tutorial are available as a part of Global Biobank Engine (<https://biobankengine.stanford.edu/degas>). Analysis scripts and notebooks are available on GitHub (https://github.com/rivas-lab/public-resources/tree/master/uk_biobank/DeGAs).

We do not use GREAT enrichment analyses to rank latent components. We applied GREAT analyses for each DeGAs component independently as described in the method section (p.16, line 655-661):

For each DeGAs component, we selected the top 5,000 variants according to their variant contribution score and performed enrichment analysis with the default parameter as described elsewhere²⁰. Since we included the non-coding variants in the analysis, we focused on GREAT binomial genomic region enrichment analysis based on the size of regulatory domain of genes and quantified the significance of enrichment in terms of binomial fold enrichment and binomial p-value. Given that we have 9,561 terms in the ontology, we set a Bonferroni-corrected p-value threshold of 5×10^{-6} .

The rank order of the DeGAs latent components, as well as the relative importance of each DeGAs components for any phenotype of interest were quantified with “phenotype squared cosine score.” Briefly, phenotype squared cosine score quantifies the relative importance of DeGAs component for any given phenotype. The full definition and their interpretation are described in the method section (p.14, line 573-583):

To quantify the contribution of the phenotypes, variants, and genes to a given component, we computed **contribution scores**. We first defined **phenotype contribution score** and **variant contribution score**. We denote phenotype contribution score and variant contribution score for some component k as $cntr_k^{phe}(i)$ and $cntr_k^{var}(j)$, respectively. They were defined by squaring the left and right singular vectors and normalizing them by Euclidean norm across phenotypes and variants:

$$cntr_k^{phe}(i) = (u_{i,k})^2$$
$$cntr_k^{var}(j) = (v_{i,k})^2$$

where, i and j denote indices for phenotype and variant, respectively. Because U and V are orthonormal, the sum of phenotype and variant contribution scores for a given component are guaranteed to be one, i.e. $\sum_i cntr_k^{phe}(i) = \sum_j cntr_k^{var}(j) = 1$.

Thanks for the opportunity to clarify these issues that are important from the user’s perspective.

(22) Large contribution of anthropometric traits

In point (15), the authors write:

“Thank you for pointing out the interesting aspects of our results. Anthropometric traits are known to be highly polygenic (Locke et al. Nature, 2015, Yengo et al. bioRxiv, 2018) and we think that is one way to interpret the results. Also, it is possible that anthropometric traits are dominating among the continuous traits with full sample size, since we didn’t include biomarkers dataset in our analysis. “

- Again, why not test these speculative comments? First, many traits — not only anthropomorphic - are known to be highly polygenic and this cannot explain the results fully. If the authors think it is a sample size issue, why not test?

Regarding point (15), let us first quote the question from reviewer 1 to clarify the context:

--- Quote from reviewer 1 starts here ---

(15) A large contribution from anthropometric traits

Is there intuition for why there is such a large contribution from anthropometric traits in the top PCs of your analysis? Is it due to strength of association? Is it a function of the phenotypic landscape of UKB?

I am also curious why the top 5 PCs are all related to different aspects of anthropometric, blood, or spirometric traits when you are using so many phenotypes for your analysis e.g. psychiatric traits. Is this a function of the enrichment of certain trait categories in UK Biobank?

--- Quote from reviewer 1 ends here ---

Thank you for the opportunity to clarify the contribution of anthropometric traits. To address your concern, we performed additional quantitative analysis and quantified the degree of polygenicity by number of independent SNP associations with the phenotype after applying a LD-clumping procedure. LD-clumping is a computational technique to characterize independent set of SNPs commonly applied to GWAS summary statistics. By comparing the polygenicity and the phenotype contribution scores for the first 5 components, we found that the phenotypes with large phenotype contribution scores tend to be highly polygenic anthropometric traits (Supplementary Figure S29, shown below):

Supplementary Figure S29. Comparison of phenotype contribution scores and the number of clumped GWAS hits for the first five DeGAs components (PC1-5). The phenotype contribution score (x-axis) and the number of clumped GWAS hits ($p < 1e-4$, y-axis) is compared. Each point is a phenotype and they are grouped and colored by phenotype categories defined in Supplementary Table S2.

We removed our previous speculative comments, included the results of this analysis as a supplementary figure (Supplementary Figure S29), and updated the main text with a reference to these additional analysis (p.8, line 324-31).

In DeGAs, we provided multiple ways to investigate the biological relevance of latent components, including quantitative scores and ontology enrichment analyses. These metrics are useful to annotate and interpret latent components, which are otherwise just mathematical objects in a high-dimensional space. For example, we found a significant contribution of anthropometric traits among the top 5 components, which reflects the pervasive polygenicity of these traits (Supplementary Fig. S29)^{38,39}. By leveraging the ability of TSVD to efficiently summarize most of the variance in the input association statistic matrix, DeGAs provides a systematic way to interpret polygenic and pleiotropic genetic architecture of common complex traits.

Thank you very much for your questions that helped us further improve our manuscript.

Response to our Reviewer 3

(23) Selection of the two PTVs for experimental follow-up

However, unfortunately the success of the approach is not proven in the last part of the manuscript. It is also not clear why the authors selected GPR115 and PDE3B for the in vitro analysis and not LIPT1 and MLPH, or two candidates from two different PC.

Thank you very much for your questions about the selection strategy of PTVs used for the experimental follow-up that helped us making the presentation clearer. We selected *GPR151* (not *GPR115*) and *PDE3B* for the follow-up experiments in adipocyte models based on the unbiased computational analyses applied on the PTVs dataset focusing on BMI, which is used as an example trait of DeGAs application throughout our manuscript.

In our computational analyses with DeGAs, we found that PC1 was identified as the top component for BMI. According to our “gene contribution scores” (Figure 4b, shown below), *PDE3B* and *GPR151* are ranked first and second with 19.03% and 12.26% of gene contribution scores, respectively, whereas *MLPH* and *LIPT1* are ranked 9 and 10 with 3.40% and 3.24% of the scores, respectively (the bar charts should be read from the bottom as is typical for that type of chart).

According to the “phenotype contribution scores” (Figure 4a, shown below), we found that first principal component (PC1) was characterized by fat-related phenotypes (with 33.8 % of the total phenotype contribution score). Thus, we selected adipocytes as the most relevant cell type for the experimental follow-up of the PC1 candidates. If we were to select candidates from a different PC, such as PC3 characterized with height-related phenotypes, we would have used another model system (in the case of PC3, probably a bone cell model) to carry out the experimental follow-up.

Hence, in summary, we picked the top PC and the top two genes contributing to this PC for functional follow-up experiments, and decided on the model system based on the nature of the phenotypes. To make our selection strategies for candidate genes and experimental models clearer, we have now added the following text to the manuscript (p.8, line 345-352):

To select candidate genes and the most relevant experimental models for the functional studies in an unbiased manner, we applied DeGAs to the PTV dataset. First, we identified PC1 as the top phenotype component contributing to BMI. Second, based on the fact that the main drivers of PC1 phenotypes were fat mass-related measurements, we chose adipocytes as our experimental model to approach candidate gene studies. Last, but not least, we selected the top two genes contributing PTVs to PC1 (GPR151 and PDE3B) and explored their functionality in fat cells, in order to illustrate the application of DeGAs computational analysis in biological research.

(24) Very low expression of *GPR151*

The data on the characterization of both genes are not convincing. *GPR151* exhibits a very low expression in adipose tissue and in 3T3-L1 and SGBS cells (with not changes during differentiation). Therefore, it is difficult to understand the effect on reduced expression of adipogenic genes like *PPARG*.

Thank you for bringing up this question. We agree with the reviewer that *GPR151* expression is low in adipose tissue and in preadipocyte models throughout adipogenic differentiation. However, we have provided evidence that the minimal expression of the endogenous *Gpr151* in preadipocytes is indispensable to initiate adipogenic process: knockdown of *Gpr151* in preadipocytes remarkably blocked adipogenic conversion (Fig. 5), while overexpression of *Gpr151* in preadipocytes did not further elevate adipogenesis (Fig. S27). The reduction of adipogenic markers, such as *Pparg*, is a reflection of the inhibited adipogenic activity in *Gpr151*-lacking cells, as we knocked down *Gpr151* in the preadipocyte stage before inducing differentiation and measured *Pparg* expression at the terminal differentiation stage to indicate the degree of differences in adipogenesis between control and *Gpr151*-deficient cells (Fig. 5d).

To clarify the potential importance of the minimal expression of endogenous *GPR151* in adipogenic precursors in imposing its effect on adipogenesis, we added the following text to the Discussion section (p.9, line 356-364):

GPCRs are known to influence adipogenesis by conveying a complex series of secondary messengers, including cAMP and calcium signals^{44,45}. The density of receptors and the timing of receptor expression during adipogenesis governs the level, timing and duration of the secondary signals, which is a critical factor in initiating and/or maintaining adipocyte conversion⁴⁴. Although the endogenous expression of GPR151 is low in preadipocytes, our results show that its presence is important to instigate the early events of adipogenic differentiation. Further investigation of the mechanism of GPR151 action will be valuable to understand its integral role in adipogenesis to the full extent.

(25) The specificity of siRNA & concern about off-target effects

The question is if the knockdown is indeed specific and if a second siRNA has been tested and if off-target effects can be excluded.

Thanks for the question and opportunity to clarify this matter. We indeed tested 3 different siRNAs targeting the same gene and observed the same downstream effects on adipogenesis. Each of these 3 siRNAs has the same intended target but contains distinct seed sequences, as well as unique potential off-target signatures. By using multiple individual siRNAs and achieving the same results, the confidence of the specificity of the siRNAs is increased and strongly imply that the observed phenotypes do indeed result from silencing the intended target. By pooling multiple siRNAs against the same target, the number and magnitude of off-target effects may be reduced due to the competition among siRNAs in the pool, while combining their on-target effects. Accordingly, we modified Figure 5c-d to show the results of 3 siRNAs both individually and together to make this clearer.

(26) Consistency between our results and IMPC KO

According to the literature, GPR115 is supposed to be involved in skin development. However, the knockout mouse generated and analyzed by the IMPC shows – similar to the experiments in cells a reduction of total fat mass. However, a large number of ko mice have an impaired fat accumulation which might have several reasons, often mediated via secondary effects.

Thank you very much for checking the external datasets. However, it should again be pointed out that we analyzed *GPR151*, not *GPR115*, in our manuscript. There is no phenotypic information is registered in IMPC on *Gpr151* (IMPC's *Gpr151* page: <https://www.mousephenotype.org/data/genes/MGI:2441887>). In fact, our study is the first experiment of a potential role of *GPR151* in fat cell development. As discussed in the text (p.9, line 389-392; quoted below), we do not exclude the possibility of other mechanisms underlying the functionality of *GPR151* in fat accumulation. The objective of the paper was to prioritize novel genes in the context of relevant phenotypes in order to initiate more effective experimental follow-up studies. For each individual gene, we believe that comprehensive and systemic studies are needed in the future to fully understand the regulatory mechanism of a gene function.

We do not exclude the contribution nor the importance of other tissues or mechanisms underlying body weight changes. Indeed, some lines of evidence support additional effects of GPR151 on obesity via the central nervous system – possibly on appetite regulation³⁷

(27) Contribution of *PDE3B* to adipocyte phenotypes

In addition, the suppression of *PDE3B* was without effect, the IMPC ko shows no effect on body weight or fat mass but a bone phenotype (long tibia).

We thank the reviewer for the chance to clarify the relevance of studying *PDE3B* in fat-related phenotypes. First, as explained in detail above, we selected the top two genes contributing to PC1 in an unbiased manner for functional experiments that were performed in fat cells due to the phenotypic profile of this PC. Then, we will respond in turn to each of the three parts of this specific question: 1) *PDE3B* knockdown had no effect in our study; 2) *Pde3b* KO in IMPC showed no effect on body weight or fat mass; 3) *Pde3b* KO in IMPC showed phenotypes in bone length.

In this study, we only tested the impact of *PDE3B* knockdown on preadipocyte differentiation and observed no effects. We cannot exclude that *PDE3B* plays a role in other functions of adipocytes in relation to energy, lipid and glucose metabolism. There is evidence that white mature adipocytes in *Pde3b* knockout mice exhibit improved mitochondrial activity and behave

as “beige” fat with more energy-burning properties (Chung, Y. W. et al., 2017). More detailed studies are needed to characterize the role of *PDE3B* in all aspects of fat biology.

In IMPC, both male and female homozygous knockout mice showed a modest increase in body weight under regular chow diet, as compared to their wild-type (WT) counterparts (IMPC *Pde3b* link:

https://www.mousephenotype.org/data/charts?accession=MGI:1333863¶meter_stable_id=MPC_BWT_008_001&&chart_type=TIME_SERIES_LINE). This phenotype in mice is consistent with our findings in humans that *PDE3B* loss-of-function variant carriers have 0.647 kg/m² higher BMI than the average of UK Biobank participants (see in the text: p. 6 line 232-234, Supplementary Figure S26). Moreover, in a study not included in IMPC database, the percentage of the visceral fat weight relative to body weight was reduced in *Pde3b* knockout mice under high-fat diet challenge, in comparison to controls. Further, a recent publication from the Million Veteran Program (MVP) indicated a beneficial association of *PDE3B* loss-of-function variation with lower triglyceride and higher HDL-cholesterol in circulation (Klarin, D. et al., 2018). All the evidence above suggests a phenotypic involvement of *PDE3B* in fat biology.

We agree with the reviewer that in IMPC, long tibia is indeed listed as a significant phenotype in *Pde3b* total-body knockout mice. In humans, our PheWAS analysis indicates that the standing height and leg fat-free mass are also driving the contribution of *PDE3B* PTVs to body weight. In this paper, we focused on studying the effect of *PDE3B* in fat cells, but did not exclude its involvement in other metabolic tissues, such as bone. In fact, fat and bone cell development are often coordinated in regulation of whole-body metabolism. All taken together, to address the reviewer’s concern over the relevance of *PDE3B* in adipocyte biology, we added and modified the following text in the Discussion (p.9, line 365-380, line 388-393):

*PDE3B, on the other hand, did not affect differentiation of preadipocytes significantly in our study. There is evidence that PDE3B plays a more notable role in differentiated mature adipocytes, the primary component of adipose tissue. As an essential enzyme that hydrolyzes both cAMP and cGMP, PDE3B is known to be predominantly expressed in tissues that are important in regulating energy homeostasis, including adipose tissue⁴⁶. White adipose tissue in *Pde3b* knockout mice behaves more as “beige” fat with improved mitochondrial activity and energy-burning properties, leading to a reduction of visceral fat mass as compared to the wild-type littermates⁴⁷. Moreover, *Pde3b* knockout in mice confers cardioprotective effects⁴⁸, and human *PDE3B* “knockout” subjects display lower circulating triglycerides and higher HDL-cholesterol in blood⁴⁹. There is a growing body of evidence that cardiometabolic health is linked to improved body fat distribution (i.e. lower visceral fat, higher subcutaneous fat)⁵⁰. Our PheWAS analysis suggests that *PDE3B* PTVs have the strongest association to hip circumference (e.g. lower-body subcutaneous adiposity) (Supplementary Fig. S24). Therefore, understanding the fat depot-specific metabolic effects of *PDE3B* may help uncover the mechanism underlying the positive relationship of *PDE3B* PTVs with peripheral fat accumulation and favorable metabolic profiles.*

In this study, we focused on evaluating the functional effects of these genes on adipocyte function and development. We do not exclude the contribution nor the importance of other tissues or mechanisms underlying body weight changes. Indeed, some lines of evidence support additional effects of GPR151 on obesity via the central nervous system – possibly on appetite regulation³⁷, while loss-of-function variant in PDE3B is also associated with height⁵¹ - another contributing factor to body weight changes.

(28) Interpretation of the experimental results from the candidate gene study

I recommend to either removing the data of the two candidates or at least to shorten it strongly and to significantly reduce the interpretation. The classification of both candidates as putative novel therapeutic targets is truly an overstatement.

Thank you for the suggestion. We shortened and modified the interpretation of the data for both genes in the Results section, only summarizing direct observations from the experiments to avoid overinterpretation. We also deleted the statement of suggesting these two genes as novel therapeutic targets in treating obesity. We agree with the reviewer that we have not performed enough experiments to fully understand the role of either gene in fat biology. We are not aiming to completely map the functional and mechanistic roles of either gene. We performed experimental studies on these two genes contributing to obesity-related traits, as unbiasedly chosen by the novel computational methods, in hopes of: 1) providing an example of how to interpret the results from DeGAs effectively and how to select candidate genes for the relevant experimental models; and 2) initiating an application of DeGAs in biological research and inspiring more state-of-art basic/translational studies about novel candidates predicted from DeGAs.

Summary of *GPR151* function in adipocytes (p.7, line 268-269, line 290-293):

These data suggest that GPR151 knockdown in adipocyte progenitor cells may block their conversion into mature adipocytes.

To sum up results from the gain- and loss-of-function studies of GPR151 in preadipocyte models, minimal but indispensable endogenous expression of GPR151 in adipose progenitor cells in generating lipid-rich adipocytes may underlie one of the mechanisms by which GPR151 promotes obesity.

Summary of *PDE3B* function in adipocytes (p.7, line 294-299):

knockdown of Pde3b in 3T3-L1 preadipocytes (Supplementary Fig. S28a) showed no significant influence on adipogenesis and lipolysis (under either basal or β -adrenergic stimulated conditions), as compared to scRNA-transfected controls (Supplementary Fig. S28b-e). Since PDE3B is expressed primarily in differentiated adipocytes (Fig. 5a-b), future research efforts should be concentrated on studying the metabolic role of PDE3B in mature adipocytes.

Summary of the value of performing the experimental studies on these two candidates (p.7, line 300-305):

Collectively, we performed functional characterization studies on the top two genes contributing to obesity-related traits, as selected based on the novel DeGAs approach in an unbiased manner, in hopes of: 1) providing an example of how to interpret the results from DeGAs effectively and how to select candidate genes for relevant experimental models; and 2) initiating an application of DeGAs in biological research and inspiring more state-of-art translational studies of novel candidates predicted from DeGAs.

Thank you very much for your time, suggestion, and comments to have substantially improved the presentation.

REVIEWERS' COMMENTS:

Reviewer #2 (Remarks to the Author):

The authors have addressed my comments.

Reviewer #3 (Remarks to the Author):

First of all I would like to appologize for mixing up GPR115 with GPR151; I'm very sorry for this mistake. Secondly, I would like to thank the authors for answering to all my questions and suggestions. I'm satisfied with the changes and Responses and recommend to publish this important set of data.

Response to the reviewers

Our responses are in blue font, the comments from reviewers are copied in black (examples are shown below):

This is an example of editorial requests or reviewer's comments
This is an example of our response.

REVIEWERS' COMMENTS:

Reviewer #2 (Remarks to the Author):

The authors have addressed my comments.

Thank you very much for your time and valuable feedback on our manuscript.

Reviewer #3 (Remarks to the Author):

First of all I would like to appologize for mixing up GPR115 with GPR151; I'm very sorry for this mistake. Secondly, I would like to thank the authors for answering to all my questions and suggestions. I'm satisfied with the changes and Responses and recommend to publish this important set of data.

Thank you very much for your time and valuable feedback on our manuscript.